# Improving the understanding of N transport in a rural catchment under Atlantic climate conditions from analysis of the concentration-discharge relationship derived from a high frequency data set

Rodríguez-Blanco, María Luz[1], Taboada-Castro, María Teresa[2], Taboada-Castro, María Mercedes[3]

[1]Physical Geography Area, History, Art and Geography Department, GEAAT Group, University of Vigo, Campus As Lagoas, 36310 Ourense, Spain
[2]Faculty of Sciences, University of A Coruña 15071 A Coruña, Spain
[3]ETSIIAA, Area of Soil Science and Soil Chemistry, University of Valladolid, 34004 Palencia, Spain

**Correspondence:** M.L. Rodríguez-Blanco (maria.luz.rodriguez.blanco@uvigo.es)

**Abstract.** Understanding processes controlling stream nutrient dynamics over time is crucial for implementing effective management strategies to prevent water quality degradation. In this respect, the study of the nutrient concentration-discharge (C-Q) relationship during individual runoff events can be a valuable tool for extrapolating the hydrochemical processes controlling nutrient fluxes in streams. This study investigated nitrogen concentration dynamics during events by analysing and interpreting the nitrogen C-Q relationship in a small Atlantic (NW Iberian Peninsula) rural catchment. To this end, nitrate ($NO_3$-N) and total Kjeldahl nitrogen (TKN) concentrations were monitored at high temporal resolution during 102 runoff events over a 6-year period. For each of the selected runoff events, C-Q response was examined visually for the presence and direction of hysteresis loops and classified into three types of responses: clockwise, anticlockwise and no hysteresis. Changes in concentration ($\Delta C$) and the hysteresis direction ($\Delta R$) were used to quantify nitrogen ($NO_3^-$ and TKN) patterns during the runoff events. The transport mechanisms varied between compounds. The most frequent hysteretic response for $NO_3^-$ was anticlockwise with enrichment. On the contrary, the main TKN dynamic was enrichment with clockwise hysteresis. Event characteristics, such as rainfall amount, peak discharge (i.e. maximum discharge of the runoff event), and event magnitude relative to the initial baseflow (i.e. the difference between the maximum discharge of the runoff event and the initial baseflow divided by initial baseflow) provided a better explanation for hysteresis direction and magnitude for TKN than antecedent conditions (antecedent precipitation and baseflow at the beginning of the event). For $NO_3^-$ hysteresis the role of hydrometeorological conditions were more complex. $NO_3^-$ hysteresis magnitude was related to the magnitude of the event relative to the initial baseflow and the time elapsed since a preceding runoff event. These findings could be used as a reference for the development of N mitigation strategy in the region.

**Keywords**: concentration-discharge, hysteresis, nitrogen, runoff events, rural catchment, Atlantic climate, NW Iberian Peninsula.

## 1 Introduction

Increasing nitrate concentrations in headwater catchments is a pressing environmental issue (EEA, 2018; Koening et al., 2021; Musolff et al., 2021). In Europe, despite the advances made in the field of improving the quality of water bodies in recent

decades, 60% of freshwater bodies fail to achieve good ecological status as established by the Water Framework Directive (Directive 2000/60/EC) (EEA, 2018). The European Directive urges Member States to monitor water quality. However, many of these countries, Spain among them, have an inadequate water monitoring network to ensure comprehensive and consistent monitoring of water bodies (EC, 2019). Historically, water quality assessments have relied on routine low-frequency monitoring at main rivers, commonly at biweekly or monthly resolution. This traditional sampling method can provide valuable information to identify sites that are under pressure due to anthropogenic activities, also to observe long-term trends in relation to land use but cannot provide knowledge on nutrient dynamics under contrasting hydrological condition (Dupas et al., 2016; Rose et al., 2018; Musolff et al., 2021), which is essential to develop suitable management programs to restore or maintain water quality (Lloyd et al., 2016; Bieroza et al., 2018).

One way of approaching the study of these dynamics requires high-frequency analysis of the nutrient concentration and discharge (C-Q) relationship at a given point in the stream during runoff events (Bieroza and Heathwaite, 2015; Lloyd et al., 2016; Rose et al., 2018; Baker and Showers, 2019). The hysteresis loop is the most typical pattern observed in the nutrient C-Q relationship (Evans and Davies, 1998). It reflects a non-linear nutrient concentration behaviour because the concentration at a given discharge on the rising limb differs from that of the same discharge on the falling limb of the hydrograph (Evans and Davies, 1998). The width, magnitude and direction of these loops have been used to investigate the sources, flow paths and transport mechanisms responsible for the export of nutrients from catchments (Evans and Davies, 1998; Butturini et al., 2008; Dupas et al., 2016; Vaughan et al., 2017; Barros et al., 2020). Hysteresis C-Q relationships can be classified into clockwise and anticlockwise according to their direction (Figure 1). Clockwise hysteresis is generally understood to reflect proximal and rapidly mobilized sources, whereas anticlockwise hysteresis reflects sources that are either proximal to the stream channel with slow transport, or those that are distant to the stream (Williams, 1989; Evans and Davies, 1998; Lloyd et al., 2016; Baker and Showers, 2019; Knapp et al., 2020). Complex hysteresis loops are often the result of the spatio-temporal variability of rainfall and antecedent moisture conditions (Ramos et al., 2015). The C-Q relationship can also result in positive or negative hysteresis slopes representing enrichment or dilution effects, respectively (Butturini et al., 2008; Lloyd et al., 2016; Vaughan et al., 2017).

**a) Clockwise hysteresis**

**b) Anticlockwise hysteresis**

N concentration (mg L$^{-1}$)

Enrichment pattern

Rising limb

Falling limb

Rising limb

Falling limb

Dilution pattern

Discharge (m$^3$ s$^{-1}$)

Enrichment pattern

Falling limb

Area

Rising limb

Rising limb

Width

Falling limb

Dilution pattern

Discharge (m$^3$ s$^{-1}$)

**Figure 1:** Idealized representation of (a) clockwise and (b) anticlockwise hysteresis loops, showing that each pattern can occur with either enrichment or dilution during the rising limb of the hydrograph. The loop width (size of the loop, i.e. the difference in concentration between the rising and falling limbs at the mid-point of discharge) and area are shown in panel b.

Numerous studies have examined the nitrogen species, particularly nitrate ($NO_3^-$) and the C-Q relationship at event scale in varying sizes of catchments under different degrees of human impact (e.g., Butturini et al., 2008; Cerro et al., 2014; Dupas et al., 2016; Outram et al., 2016; Vaughan et al., 2017; Baker and Showers, 2019; Musolff et al., 2021; Winter et al., 2021), showing evidence of diverging C-Q relationships in agricultural and forest catchments. For example, in intense agricultural management catchments $NO_3^-$ showed a consistent response between events, dominated by clockwise enrichment or dilution patterns (Cerro et al., 2014; Dupas et al., 2016; Outram et al., 2016). In the case of forested catchments, where the $NO_3^-$ response was highly variable, an $NO_3^-$ enrichment C-Q pattern with clockwise or anticlockwise hysteresis was prevalent (Vaughan et al., 2017; Musolff et al., 2021). Hysteresis patterns can change from one runoff event to another and several factors such as antecedent wetness conditions, rainfall depth, and runoff volume, as well as event-water contributions, play an important role in controlling the variability in the $NO_3^-$ C-Q relationship among various hydrological events (Outram et al., 2016; Baker and Showers, 2019; Knapp et al., 2020). For example, Knapp et al. (2020) observed dilution behaviour during wetter conditions and stronger mobilization following drier ones and during events with larger event-water contributions. However, others found that $NO_3^-$ hysteresis behaviour was better explained by runoff event magnitude and rainfall intensity (Butturini et al., 2008; Aguilera and Melack, 2018) . Comparatively, approaches for the analysis of organic N C-Q relationships at event scale are scarce and when the focus is mostly on organic dissolved nitrogen (e.g., Kaushal and Lewis, 2003; Chen et al., 2012; D'Amario et al., 2021). The literature suggests that, even though dissolved organic nitrogen accounts for only part of the nitrogen in streams draining in agricultural management catchments, both particulate and dissolved forms of organic nitrogen can constitute a substantial quantity of the total nitrogen export in rural and forested catchments (Hagedorn et al., 2000; Kaushal and Lewis, 2003; Aguilera and Melack,2018). Thus, (Lorite-Herrera et al., 2009)pointed out that dissolved organic nitrogen is the predominant form of nitrogen in an intensively farmed catchment in southeastern Spain (72-97%).

Similarly, Hagedorn et al. (2000) and Kaushall and Lewis (2003) found that organic nitrogen accounted for approximately 60% of total annual nitrogen in their study areas: the Erlenbach headwaters basin (Switzerland).and the Rocky streams (Canada).(Bernal et al., 2005), on the other hand, reported a moderate fraction (35%) of the N annual flux in the form of dissolved organic nitrogen in a Mediterranean forested catchment in Catalonia (NE Spain), while Rodríguez-Blanco et al. (2015) reported a more than 25% contribution of total Kjeldahl nitrogen to total nitrogen in a rural catchment in NW Spain. Therefore, the analysis of $NO_3^-$ and total organic nitrogen C-Q relationships may provide useful information about the processes regulating N transport through the landscape. So far, there are few studies integrating both $NO_3^-$ and total organic nitrogen C-Q relationships into a coherent framework in the southwest of Europe, partly due to the limited availability of high-frequency data. This information is essential in order to anticipate changes in the quality of freshwater resources in compliance with the Water Framework Directive planning and monitoring norms. Therefore, it is necessary to provide new information on the issue to augment current studies across Europe and the Iberian Peninsula, in particular.

In this context, the aim of this study was to understand differences in the behaviour of $NO_3^-$ and total Kjeldahl nitrogen (TKN) concentrations. For this, we used high-frequency measurements of $NO_3^-$ (expressed as $NO_3$-N) and TKN concentration obtained during runoff events of contrasting magnitudes at the outlet of an Atlantic headwater catchment located in the NW Iberian Peninsula. More specifically, the study aims to explore questions such as i) how $NO_3^-$ C-Q behaviour differs from that of TKN C-Q; ii) how variable C-Q relationships between individual events are; and iii) whether variability in C-Q relationships can be explained by specific rainfall-runoff event characteristics, such as antecedent wetness conditions, rainfall, runoff volume, etc. The selected catchment (Corbeira, 16 km$^2$, NW Iberian Peninsula) is of particular interest, as it is a tributary of the Mero River, which discharges into the Abegondo-Cecebre reservoir - the main water supply for the city of A Coruña and surrounding municipalities (450 000 inhabitants) - and finally drains into the Atlantic Ocean through the ria of O Burgo. The Cecebre-Abegondo reservoir is a Natural 2000 EU site, classified as a Special Area of Conservation (ES1110004) in 2014 under the EU Habitats Directive (Directive 92/43/ECC) and one of the Core Zones of the Mariñas Coruñesas e Terras do Mandeo Biosphere Reserve, sustaining important bird, macroinvertebrate, and fish populations. Nevertheless, the ecological status of the Abegondo-Cecebre reservoir has deteriorated in the last few decades due to pollution, the presence of invasive alien species and fluctuations of river flow discharge (Ameijenda et al., 2010).

## 2 Material and methods

### 2.1 Study site

The study was conducted in a headwater catchment of 16 km$^2$ located in NW Spain, approximately 30 km southeast of the city of A Coruña (Galicia, NW Iberian Peninsula) (Figure 2). The catchment is characterized by low drainage density (1.38 km km$^{-2}$) a mean slope of 19% and a maximum altimetric amplitude of 410 m (65-675 m). The bedrock consists of basic schist of the Órdenes Complex (IGME, 1981) and the soils are predominantly Umbrisols and Cambisols (IUSS, 2015), with a silt and silty-loam texture, variable organic matter content (4.4-10.5%) and acid pH in the surface soil layer. The soils have a high

infiltration capacity, so overland flow is unusual. Groundwater is the dominant source of water to the stream and the baseflow

index is 0.75 (Rodríguez-Blanco, et al., 2012). The catchment land cover comprises a mixture of forest (65%), agricultural fields (30%) and impervious areas (5%), consisting of roads and single-family homes that are not always connected to sewage disposal systems. Agricultural areas are dominated by pastures (26% of total area), the remaining fields (4%) grow maize and winter cereals. Organic and inorganic fertilizers are commonly used in agricultural areas throughout the year, including the wettest months. Forest areas are not fertilized. The annual N input to the Corbeira catchment is approximately 37.8 kg N ha$^{-1}$

(Rodríguez-Blanco et al., 2015), indicating relatively low nitrogen inputs in the Corbeira catchment compared to catchments with more intensive agriculture.

The study area is located within the Eurosiberian biogeographic region, particularly in the Cantabrian-Atlantic province (Instituto Geográfico Nacional, 2008). It is included in the temperate oceanic climate region (Csb) according to Köppen-Geiger classification. Mean annual rainfall and temperature for the period 1983-2020 are 1075 mm and 13ºC, respectively (data from

125 the station ID 10045 of the official meteorological service of the Galician Government-Meteogalicia, located approximately 8 km from the catchment outlet). The wettest period is from October to March, and the driest and hottest months are usually in summer (June-September). The hydrological regime is pluvial oceanic, with maximum discharge in December and low flows from June to September. Mean daily-recorded discharge is 0.18 m$^3$ s$^{-1}$. For more detailed information of the hydrological behaviour of this catchment see (Rodríguez-Blanco et al., 2012; 2020).

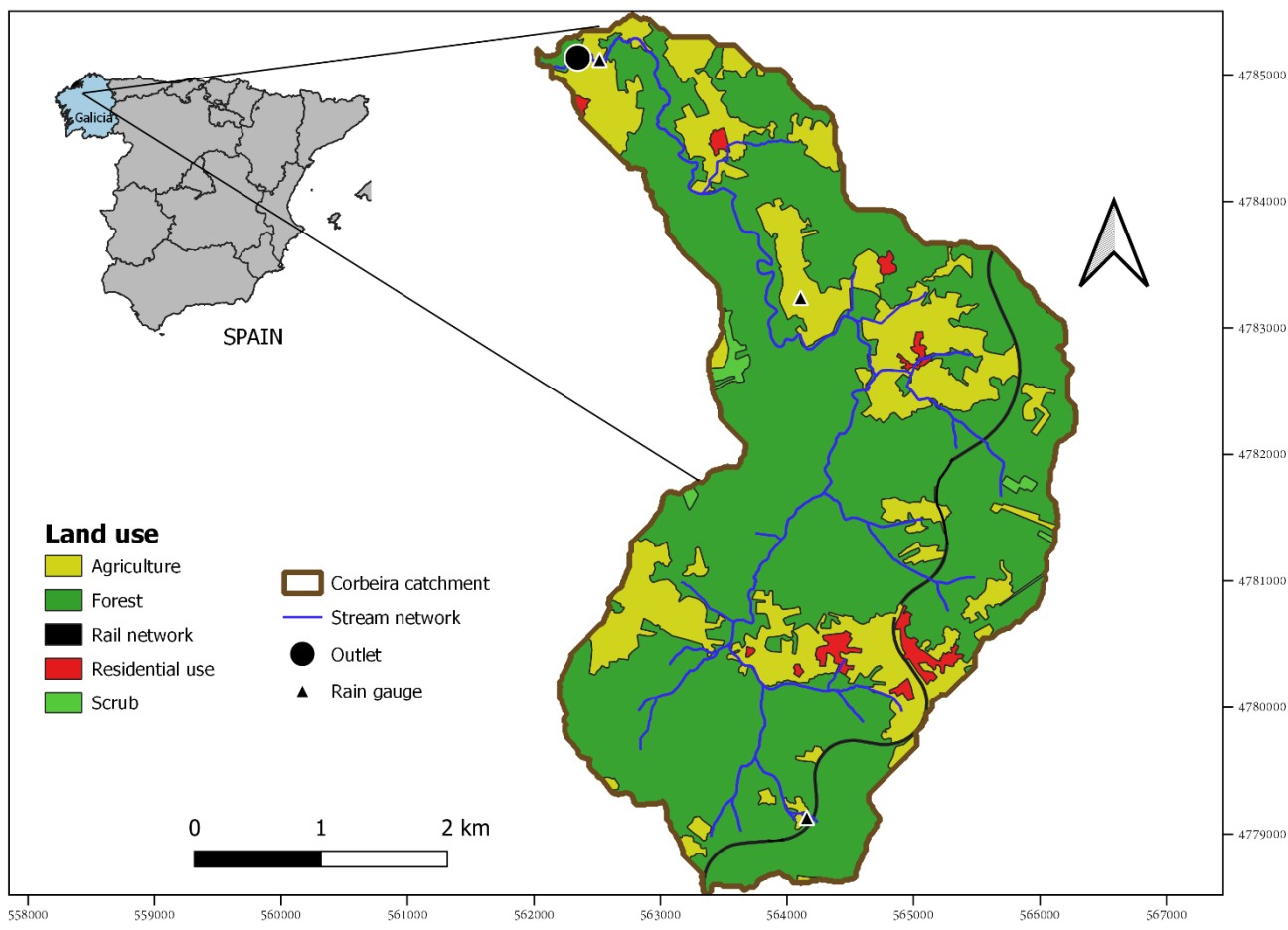

**Figure 2:** Location and land use of the Corbeira catchment.

## 2.2 Data acquisition: monitoring, sampling, and water analysis

The research period comprised six hydrological years (1$^{st}$ October to 30$^{th}$ September), during which rainfall, discharge, and N (NO$_3^-$ and TKN) concentrations were measured. Rainfall was monitored at 10-min intervals using three gauges (0.2 mm resolution) distributed across the catchment. The Thiessen Polygon method (Linsey et al., 1949) was used to calculate the mean rainfall in the catchment. Water discharge was obtained at 10-min resolution at the catchment outlet. Stream water level was measured with a differential pressure transducer sensor (ISCO-720) coupled to an automatic water sampler (ISCO 6712-FS) at 1 min intervals and recorded at 10-min resolution. The water level was then converted into discharge by rating-curve development over a wide range of discharge conditions at the sampling location.

Stream water samples were taken at the catchment outlet during runoff events using the automatic water sampler (Teledyne ISCO, Portable Sampler 6712-FS) fitted with 24 polypropylene 1-litre bottles. The pump inlet of the autosampler was placed near the pressure sensor. In order to characterise the dynamics of N concentrations from the onset of runoff events, the sampler was configured to start collecting water samples when the stream water level rose by 2-3 cm above that at the beginning of a rainfall event. The criteria established by Dunkerley (2008), i.e., considering the minimum inter-event time and the minimum rainfall depth, have been used to define rainfall events. In this assessment, a single rainfall event was defined for a minimum rainfall depth of 5 mm (not interrupted by gaps of more than 1 h) and a minimum inter-event time of 10 h. The end of a rainfall event was determined by the last non-zero rainfall pulse. The choice of these thresholds to define a single rainfall event is due to the hydrological response to less than 5 mm of rain volume is practically undetectable in the study area (Rodríguez-Blanco et al., 2012; Palleiro et al., 2014). In addition, the 10 h inter-event period (a value of more than double the concentration time of the catchment; 4.5 h; (Rodríguez-Blanco et al., 2012) was selected to ensure that the hydrological response of the catchment to individual rainfall events did not overlap between events.

Water samples were taken at fixed intervals (1-8 h) during the rising and falling limbs of the hydrograph to collect key runoff phases. The pumping frequency was modified manually depending on the expected magnitude and duration of runoff events based on weather forecasts together with the experience accumulated from water sampling over the years. Samples were removed from the autosampler within a few hours after runoff events and transported to the laboratory where they were stored in the dark and refrigerated at 4ºC until the total Kjeldahl nitrogen (TKN) and nitrate (expressed as $NO_3$-N) concentrations were analysed. TKN concentrations were determined by Kjeldahl digestion of unfiltered samples following the APHA method (APHA, 1998), whereas $NO_3^-$ concentrations were analysed by capillary electrophoresis after sample filtration (0.45 μm).

**2.3 Selection of runoff events and description of C-Q hysteresis**

In this study, the runoff events were identified as any hydrological response to rainfall where the discharge increased by at least 1.5 times of that recorded at the beginning of the event (Tardy, 1986; García-Ruiz et al., 2005). For each runoff event, the stream discharge was separated into two components (direct runoff and baseflow) using the constant slope hydrograph separation method by Hewlett and Hibbert (1967) and a constant slope of 1.83 L $s^{-1}$ $km^{-2}$ $d^{-1}$, as suggested in other Spanish forested catchments (e.g., Latron et al., 2008). The starting point of the runoff event has been identified as a sudden increase in discharge in response to a rainfall, meaning that stream discharge increases by at least 5% in a 30-min interval (three-time steps). The end of the event was fixed at when direct runoff ended (i.e. the time when the constant slope line used in the separation of runoff components intersects the falling limb of the hydrograph) (Figure 3) or when a different hydrological event, identified by another increase in stream discharge (at least 5 % in three time steps) following the associated rainfall, commenced. In the latter case, the limit between runoff events was defined as the minimum discharge recorded between them. Potential mechanism and processes controlling the hysteresis dynamics can be represented and quantified by potential explanatory variables that may explain the variability of hysteresis (Fovet et al., 2018). Two categories of potential explanatory

variables can be considered: catchment characteristic (slope, soil type, land use) and event-scale hydro-biogeochemical variables. Within the same catchment the variability of hysteresis patterns are highly influenced by event-scale properties ranging from hydrological, biogeochemical, and antecedent conditions (Butturini et al., 2006; Outram et al., 2016; Aguilera and Melack, 2018; Heathwaite and Bieroza, 2020). In order to study the influence of different event-scale hydro-biogeochemical variables on hysteresis dynamics of $NO_3^-$ and TKN the events were characterized by three groups of variables i) those related to antecedent wetness conditions (i.e. variables characterizing the conditions prior to the event), ii) event variables (rainfall and discharge) and iii) variables related to $NO_3^-$ and TKN concentrations (Table 1) as the nitrogen dynamic (response variables-hysteresis descriptors). Antecedent wetness conditions were described by accumulated rainfall 7 and 15 days prior to the event (AP7d, and AP15d, respectively, mm), the discharge at the beginning of the event ($Q_b$, $m^3$ $s^{-1}$) and the time elapsed from the previous runoff event ($\Delta t$, h). Event variables included rainfall amount (P, mm); maximum 10-min rainfall intensity (IP10, mm $h^{-1}$); rainfall kinetic energy (KE, MJ $ha^{-1}$) determined according to Wischmeier and Smith (1958); peak discharge ($Q_{max}$, $m^3$ $s^{-1}$); total runoff (TR, mm), i.e. total water volume during the runoff event (direct runoff +baseflow); magnitude of the event relative to the initial baseflow ($\Delta Q$; i.e. $(Qmax-Q_b)/Q_b*100$, %); relative length of the rising limb (RL, %) given by $RL=R_D/S_D*100$ where $R_D$ and $S_D$ are the length (h) of the rising limb of the hydrograph (i.e. the part of a hydrograph from the start of the runoff event to the peak discharge) and of the entire hydrograph (i.e. runoff event duration, i.e. the time difference in hours between the start and finish), respectively; slope of the initial phase (2 h) of the hydrograph falling limb (i.e. the part of the hydrograph from peak discharge to the end of the runoff event) estimated using an exponential model (Singh, 1988) (K, 1/day). Finally, to describe $NO_3^-$ ($NO_3$-N) and TKN concentrations, the initial, maximum, and discharge-weighted mean concentrations of $NO_3^-$ and TKN measured during the events were included ($NO_3$-$NC_{initial}$, $NO_3$-$NC_{max}$, $NO_3$-$NC_{mean}$, $TKNC_{initial}$, $TKNC_{max}$, $TKNC_{mean}$, respectively; mg $L^{-1}$). The discharge-weighted mean concentration of the event was computed as total load divided by the total flow.

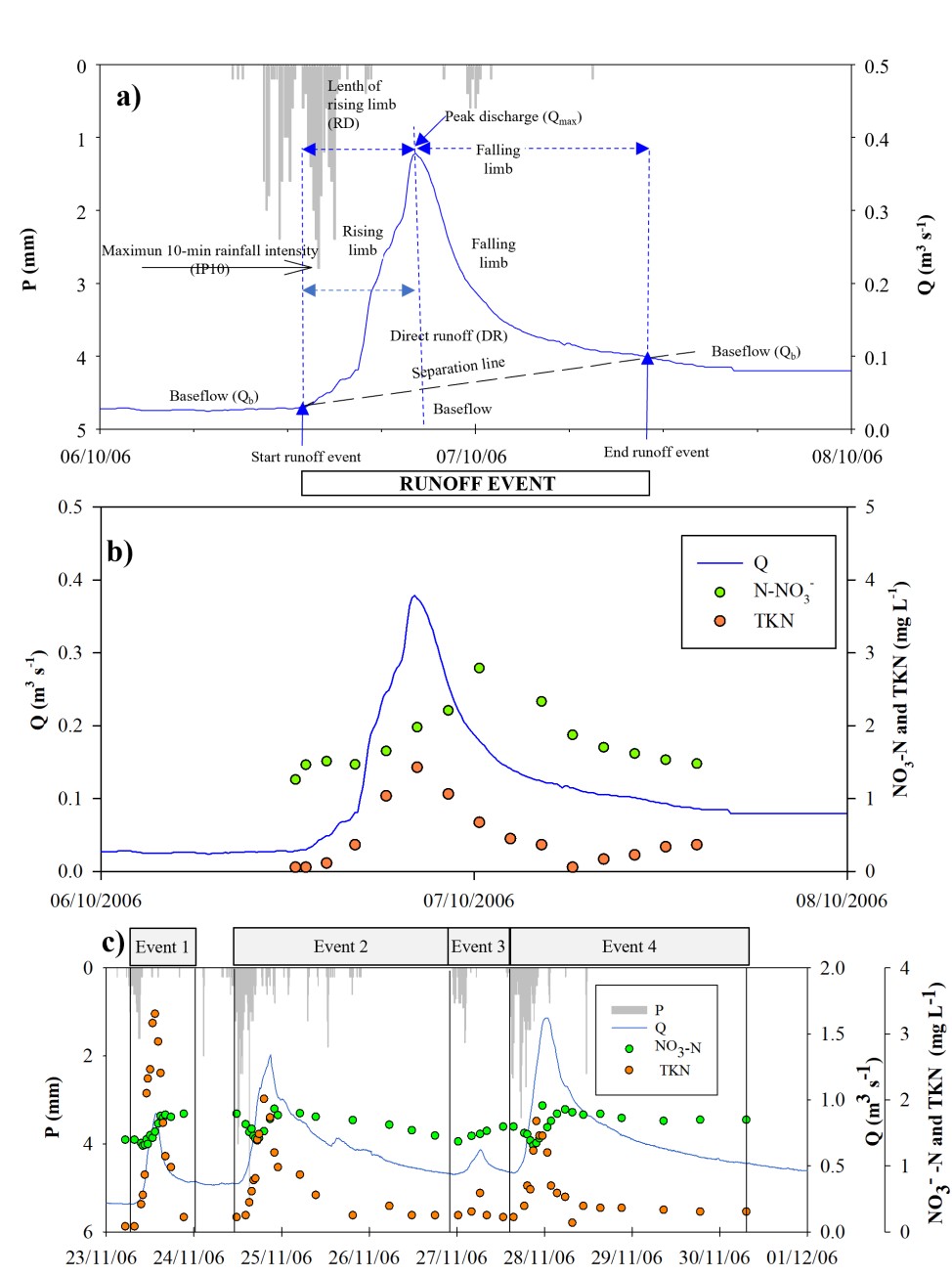

**Figure 3:** a) Diagram illustrating the hydrograph separation in two components (direct runoff and baseflow) using the constant slope hydrograph separation by Hewlett and Hibbert (1967). Some variable characteristics of the rainfall-runoff events are also indicated. b) Example of evolution of $NO_3^-$ ($NO_3$-N) and TKN concentrations during a runoff event. c) Sequence of rainfall-runoff events and evolution of $NO_3^-$ ($NO_3$-N) and TKN concentrations.

During the entire monitoring period, 173 runoff events were identified; 156 were sampled, and the other 17 were missed because of technical problems with the equipment. In this assessment, the only events used were those with a single peak discharge and at least two samples collected on each limb of the hydrograph and one at or close to peak discharge (30 min), the minimum needed to identify rotational direction (Evans and Davies, 1998), and the rest were discarded. By applying these criteria, only 102 of the 156 events sampled were included in the C-Q.

For each of the selected runoff events, C-Q $NO_3^-$ and TKN response was evaluated visually for the presence and direction of a hysteresis loop (by plotting concentration versus discharge) and grouped into three types of responses: clockwise, anticlockwise and no hysteresis. Events with a "figure-eight" hysteresis pattern were classed as a hysteresis response, with the direction depending on the succession of the peak concentration and peak discharge (i.e. clockwise, or anticlockwise), similarly to Bieroza and Heathwaite (2015).

**Table 1.** Characteristics of the runoff events (n=102) selected during the study. CV: Coefficient of variation.

| Variable | | Mean | Minimum | Maximum | CV (%) |
|---|---|---|---|---|---|
| **Antecedent conditions** | Accumulated rainfall 7 days before the event (AP7d, mm) | 35.18 | 0.60 | 124.40 | 81 |
| | Accumulated rainfall 15 days before the event (AP15d, mm) | 67.29 | 1.00 | 222.10 | 77 |
| | Discharge at the beginning of the event ($Q_b$, $m^3\ s^{-1}$) | 0.21 | 0.03 | 0.64 | 60 |
| | Time from the previous runoff event ($\Delta t$, h) | 236.83 | 0.00 | 4065.02 | 212 |
| **Event conditions** | Rainfall amount (P, mm) | 22.24 | 5.00 | 74.40 | 69 |
| | Maximum 10-min rainfall intensity (IP10, mm $h^{-1}$) | 2.35 | 0.40 | 9.20 | 71 |
| | Rainfall kinetic energy (KE, MJ $ha^{-1}$) | 3.16 | 0.52 | 10.49 | 74 |
| | Peak discharge ($Q_{max}$, $m^3\ s^{-1}$) | 0.49 | 0.10 | 1.62 | 65 |
| | Total runoff volume (TR, mm) | 2.54 | 0.26 | 1.88 | 99 |
| | Magnitude of the event relative to initial baseflow ($\Delta Q$, %) | 165.57 | 17.65 | 853.33 | 92 |
| | Relative length of the rising limb (RL, %) | 33.63 | 11.63 | 64.35 | 38 |
| | Slope of the initial phase of the hydrograph falling limb (K, 1/day) | -0.016 | -0.053 | -0.001 | 72 |
| | Runoff event duration ($S_D$, h) | 32.41 | 9.80 | 115.80 | 59 |
| **$NO_3$ and TKN concentrations during the events** | *$NO_3^-$* | | | | |
| | Initial concentration ($NO_3$-$NC_{initial}$, mg $L^{-1}$) | 1.22 | 0.70 | 2.84 | 27 |
| | Maximum concentration ($NO_3$-$NC_{max}$, mg $L^{-1}$) | 1.60 | 0.71 | 5.09 | 41 |
| | Mean concentration ($NO_3$-$NC_{mean}$, mg $L^{-1}$) | 1.31 | 0.70 | 2.27 | 25 |
| | *TKN* | | | | |
| | Initial concentration ($TKNC_{initial}$, mg $L^{-1}$) | 0.25 | 0.01 | 2.55 | 129 |
| | Maximum concentration ($TKNC_{max}$, mg $L^{-1}$) | 1.47 | 0.08 | 9.41 | 96 |
| | Mean concentration ($TKNC_{mean}$, mg $L^{-1}$) | 0.6375 | 0.04 | 2.88 | 79 |

| | | | | | |
|---|---|---|---|---|---|
| **Hysteresis descriptors** | *NO₃⁻* | | | | |
| | Hysteresis direction (ΔR, %) | -20.62 | -93.00 | 60.00 | 151 |
| | Hysteresis magnitude (ΔC, %) | 3.86 | -44.28 | 47.20 | 395 |
| | *TKN* | | | | |
| | Hysteresis direction (ΔR, %) | 4.78 | -72.00 | 69.00 | 513 |
| | Hysteresis magnitude (ΔC, %) | 66.15 | -70.35 | 98.45 | 61 |

Following the methodology proposed by Butturini et al. (2008) the form, rotational patterns, and slope of $NO_3^-$ and TKN hysteresis loops were characterized by two descriptors: hysteresis magnitude (ΔC, %) and hysteresis direction (ΔR, %). Hysteresis magnitude (ΔC) describes the relative changes in nitrogen ($NO_3^-$ or TKN) concentration and hysteresis slope, and is calculated using the following equation:

$$\Delta C \begin{cases} \frac{C_s - C_b}{C_{max}} * 100 \text{ if } C_s > C_b \\ \frac{C_s - C_b}{C_b} * 100 \text{ if } C_s < C_b \end{cases} \tag{2}$$

where $C_s$ and $C_b$ are the nitrogen ($NO_3^-$ or TKN) concentrations at peak discharge and baseflow, respectively, and $C_{max}$ is the highest concentration measured in the stream during the runoff event. ΔC ranges from - 100 to 100%, where positive values indicate hysteresis loops following a positive slope with respect to the discharge, i.e. element flushing, and negative values indicate the opposite, i.e. solute dilution. Hysteresis direction (ΔR) reflects the entire element dynamics during runoff events and provides information on the area (magnitude) and rotational (direction) pattern of the C-Q hysteresis. ΔR is calculated by the following equation:

$$\Delta R = R * Ah * 100 \tag{3}$$

where Ah is the area of the C-Q hysteresis, estimated after standardizing discharges and concentrations to unit on dividing nitrogen ($NO_3^-$ or TKN) concentrations and discharge by their peak values and calculating the area under the loop. Thus, Ah takes values between 0 and 1; as the Ah value approaches zero, the loop becomes more linear, indicating minor differences in nitrogen ($NO_3^-$ or TKN) concentrations between the rising and falling limbs. On the contrary, an Ah value close to 1 indicates that the area of the hysteresis loop is large, showing strong differences in nitrogen ($NO_3^-$ or TKN) concentrations in both limbs of the hydrograph at similar discharge. R describes the rotational pattern of the hysteresis. If the C-Q hysteresis is clockwise, R=1, and if it is anticlockwise, R=-1; for ambiguous or non-existent hysteresis, R=0. The descriptor ΔR also varies from -100 to 100 %.

The variability of $NO_3^-$ and TKN hysteresis descriptors was examined by plotting hysteresis magnitude (ΔC) versus hysteresis direction (ΔR). The plots can be divided in 4 zones (Butturini et al., 2006), each of which identifies a C-Q response type

(Figure 4). For this, ΔC and ΔR descriptors were divided into two categories ("-1", "1"): $\Delta C < 0$ (element dilution); $\Delta C > 0$
(element release); $\Delta R < 0$ (anticlockwise loop); $\Delta R > 0$ (clockwise loop).

## 2.4 Statistical methods

To assess the main links between hysteresis descriptors (response variables) and the different hydro-meteorological and
biogeochemical (i.e. $NO_3$ and TKN concentrations) variables (explanatory variables), standard statistical methods were used,
such as correlation (Pearson correlation coefficient) and a redundancy analysis (RDA). A Pearson correlation analysis provides
a quantitative estimation of the degree of linear correlation between hydro-meteorological and biogeochemical variables with
hysteresis descriptors. The RDA was used to investigate the multivariate relationship between response variables (hysteresis
descriptors) and explanatory variables (hydro-meteorological and biogeochemical variables). RDA is an extension of principal
component analysis to datasets where there are multiple response variables modelling the effects of explanatory variables on
response variables assuming linear relationships, thus it enables an examination of the variation in a matrix of variables
(hysteresis descriptors) that can be explained by a matrix of predictor/explanatory variables (hydro-meteorological and
biogeochemical variable) (Legendre and Legendre, 2012). Moreover, RDA can be used when variables cannot be considered
strictly independent from each other, as in this case (ΔC and ΔR), where both descriptors are undoubtedly related. The RDA
output was represented in a biplot graph showing the correlation between explanatory and response variables given by the
cosine of the angle between vectors. Thus, vectors pointing in roughly the same direction represent a positive correlation, those
pointing in opposite directions show a negative correlation and vectors crossing at right angles indicate a near zero correlation.

## 3 Results

### 3.1 Average characteristics of the rainfall-runoff events

The main characteristics of the selected events for this study are summarized in Table 1 and Fig. 4. High variability in the
variables defining the events was observed. Thus, the events varied greatly in terms of antecedent conditions (AP7d:
0.60 - 124.40 mm, AP15d: 1.00 - 222.10 mm, $Q_b$: 0.03 - 0.64 $m^3$ $s^{-1}$), meteorological (P: 4.00 -74.40 mm, KE: 0.52 - 10.49
MJ $ha^{-1}$) and hydrological features ($Q_{max}$: 0.10 - 1.62 $m^3$ $s^{-1}$, ΔQ: 17.65 - 853.33%, $S_D$: 9.80 - 115.80 h), showing that those
selected cover a wide range of meteorological and hydrological conditions. These events (i.e. the 102 used in the study) can
be considered representative of the rainfall-runoff event characteristics of the study period, because the meteorological and
hydrological data used in this study are within the 5th to 95th percentile range of rainfall, antecedent rainfall and discharge of
the events occurring in the area during the study period.

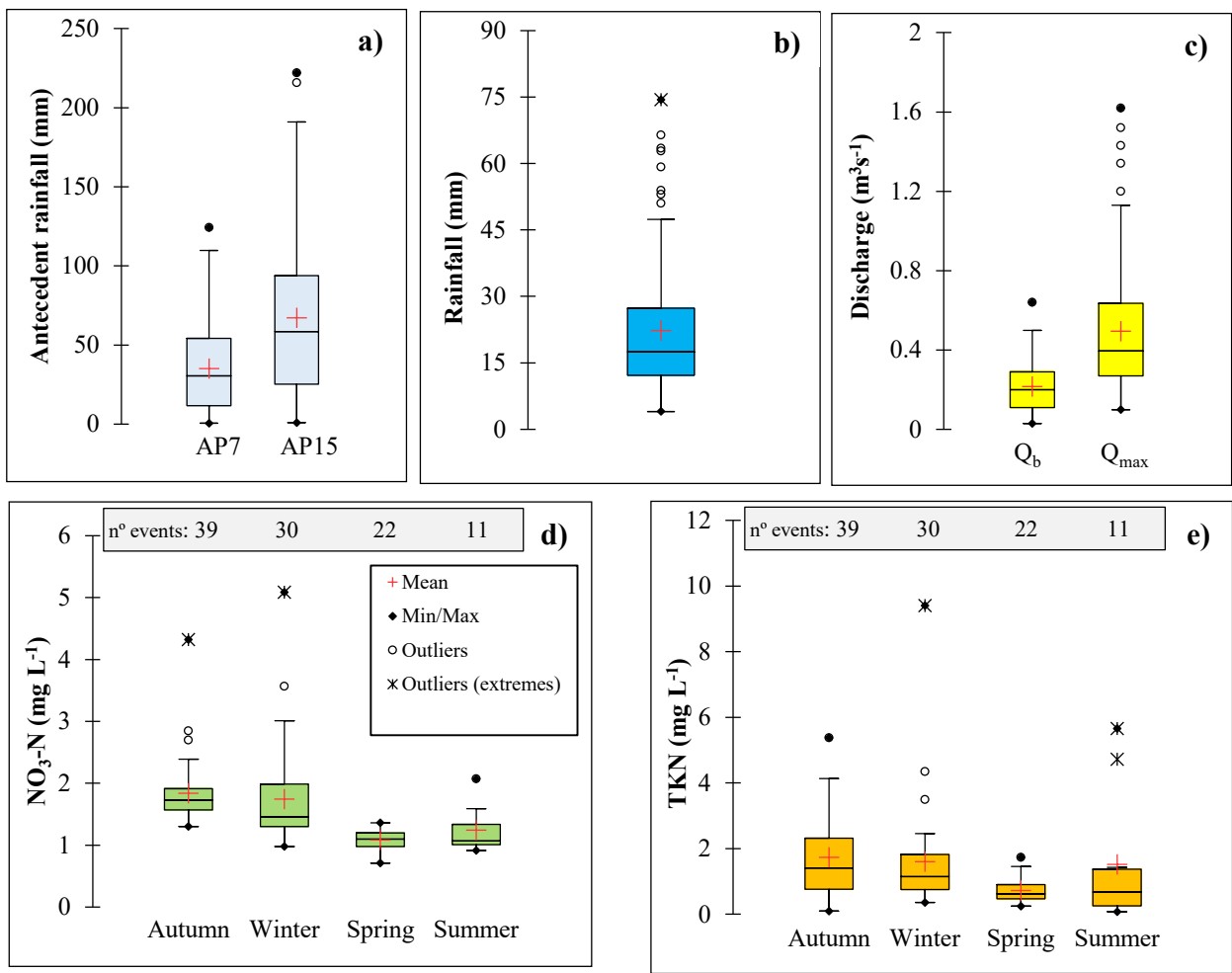

**Figure 4**: Box plot summarizing the characteristics of the events: (a) antecedent rainfall, (b) rainfall amount, (c) discharge, (d) NO₃ concentrations (NO₃-N) and (d) TKN concentrations. Box: interquartile range defined as the difference between the 3$^{rd}$ and the 1$^{st}$ quartile (interquartile range-IQR); horizontal line within the box: median; whiskers: 1.5 times interquartile range beyond the box, outliers: values between 1.5 and 3 times the IQR larger than 3$^{rd}$ quartile or between 1.5 and 3 times the IQR lower than the 1$^{st}$ quartile; outlier (extremes): values 3 times the IQR larger than 3$^{rd}$ quartile or 3 times the IQR lower than the 1$^{st}$ quartile.

From the selected 102 rainfall-runoff events, 39 occurred in autumn (October, November, and December), 30 in winter (January, February, and March), 22 in spring (April, May, and June) and 11 in summer (July, August, and September), so that about 70% were concentrated in the wettest period of the year (October-March). The magnitude of the runoff events tended to be high in autumn and winter when soil moisture is high, while in summer, when the catchment is dryer, the event magnitude tended to be lower (Rodríguez-Blanco et al., 2012) (Figure 5c). In the study area, the runoff events are usually linked to low-volume (mean P = 22.24 mm) and intensity (mean IP10 = 2.35 mm h$^{-1}$) rainfall events of long duration (mean = 14.8 h, min = 2 h, max = 52.5 h), although several events with high volume (P > 50 mm) and intensity (IP 10 = 9.1 mm h$^{-1}$) rainfall were

registered during the study (Table 1). For most runoff events, an increase in NO$_3$-N and TKN concentrations with discharge were observed, but the magnitude of the increase varied markedly from one event to another. The mean and maximum N (NO$_3$-N and TKN) concentrations also varied among runoff events, especially for TKN; the maximum TKNC$_{mean}$ and TKNC$_{max}$ values were two orders of magnitude higher than the respective minimum values (Table 1). The highest values of both elements were recorded during autumn and winter events (Figure 4).

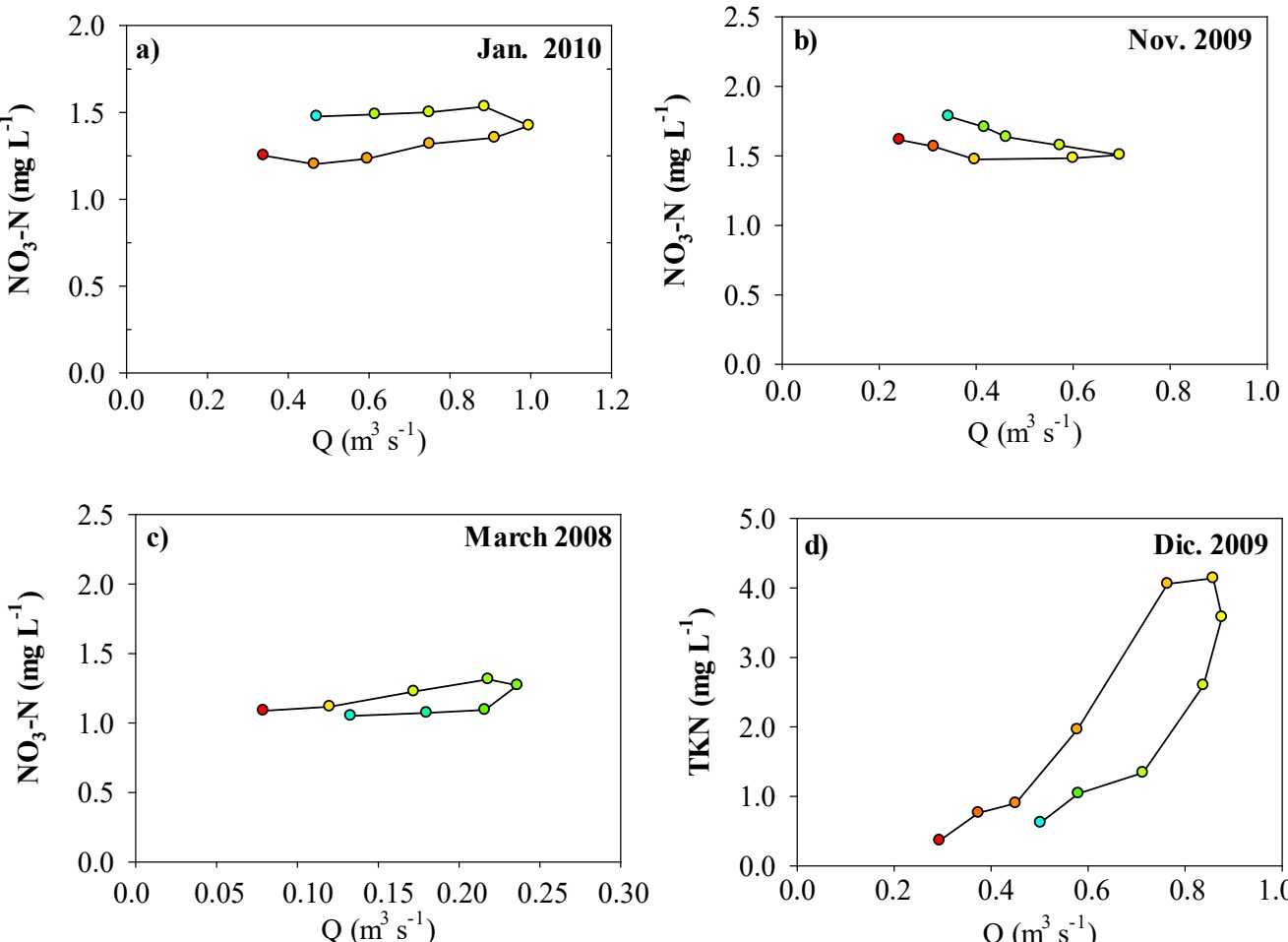

**Figure 5**: Examples of different types of NO$_3^-$ (a, b and c) and TKN (d) hysteresis patterns observed in the Corbeira catchment during the monitoring period, where red indicates the start and blue the end of the runoff events. (a) Enrichment with anticlockwise pattern, (b) Dilution with anticlockwise pattern, (c and d) Enrichment with clockwise pattern Circular arrows show the direction of clockwise and anticlockwise hysteresis.

### 3.2 Hysteresis direction and magnitude

The study of the relationship between the N (NO$_3$-N and TKN) concentration and discharge revealed different hysteresis patterns for both elements in the catchment (Figure 5 and 6). For NO$_3^-$, the parameter describing the change in concentration during the runoff events returned positive values ($\Delta C \geq 0\%$) in 63% of the events. These positive values show that NO$_3$-N concentrations during the runoff events were mostly greater than before the event; but 34% had $\Delta C$ values between 0% and 10%, indicating a small increase in NO$_3$-N concentrations (Butturini et al., 2008). Based on hysteresis classification, 74% of the events exhibited anticlockwise hysteresis ($\Delta R < 0$), 21% clockwise hysteresis ($\Delta R > 0$) and the remaining 5% showed no or unclear hysteresis patterns ($\Delta R = 0$). However, it should be noted that approximately 13% of events returned $\Delta R$ values between -10% and 10%; therefore, the hysteresis area is considered to be small. NO$_3^-$ data are in all areas in the $\Delta C$ *vs.* $\Delta R$ unit plane (Figure 6 top), although the hysteresis loops are located mainly in zones D and C (Figure 6 top), indicating dilution (negative $\Delta C$) or flushing (positive $\Delta C$) and anticlockwise hysteresis loops (negative $\Delta R$).

TKN concentrations increased in almost all runoff events compared with pre-event values (positive $\Delta C$ in 93% of events), indicating that TKN flushing clearly predominates over dilution. In fact, the parameter describing the change in concentration during runoff events ($\Delta C$) presented negative values in only 7% of cases (Figure 6 bottom), all of which were characterized by low rainfall. The rotational patterns of the TKN-Q hysteresis ranged from clockwise ($\Delta R > 0$) to anticlockwise ($\Delta R < 0$) (Figure 6 bottom). About 53% of the events showed clockwise hysteresis, 39% anticlockwise hysteresis and the remaining 8% showed no or unclear hysteresis patterns; although it should be noted that 29% of the events showed small areas of the hysteresis loop ($\Delta R$ values stood between -10% and 10%). The hysteresis loops are located mainly in zones A and D (Figure 6 bottom), suggesting a flushing (positive $\Delta C$) and clockwise (positive $\Delta R$) or anticlockwise loops (negative $\Delta R$).

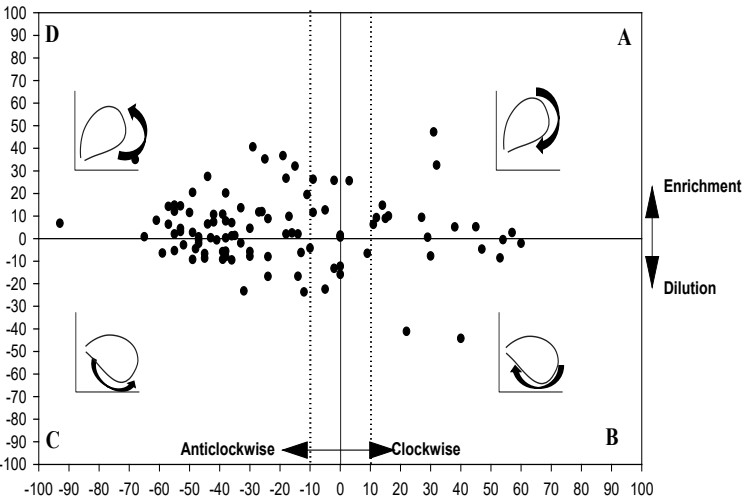

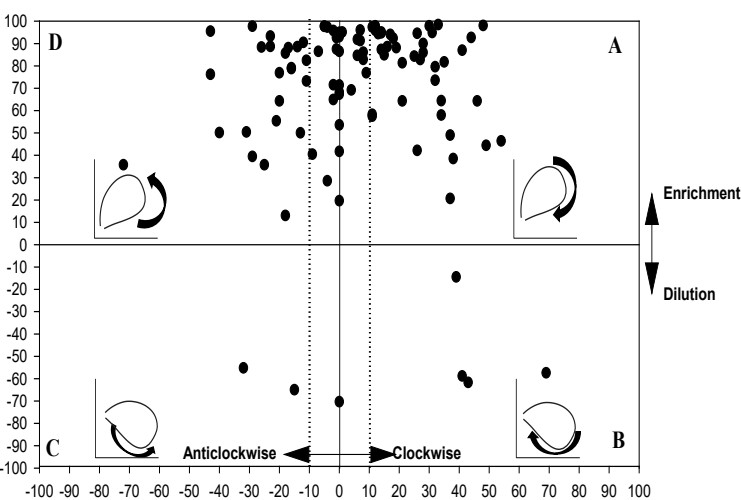

**Figure 6**: Representation of the C-Q hysteresis characteristics (ΔR, ΔC) of NO$_3^-$ (up) and TKN (bottom) in the unity plane (A: clockwise hysteresis with enrichment, B: clockwise hysteresis with dilution, C: anticlockwise hysteresis with dilution, D: anticlockwise hysteresis with enrichment). The vertical and horizontal dotted lines delimit hysteresis loops with a small area. Circular arrows show the direction of clockwise and anticlockwise hysteresis (authors' own creation based on Butturini et al.

(2006).

### 3.3 C-Q hysteresis response controls

The relationships between hysteresis descriptors and hydrological and biogeochemical variables were analysed using a Pearson correlation matrix and an RDA analysis (Table 2 and Figure 7) in order to identify the variables that might explain C-Q hysteresis patterns. The results of the correlation analysis showed that the hysteresis direction and magnitude were more closely related to certain event characteristics than antecedent conditions (Table 2). Thus, of the representative variables of the event antecedent conditions, relevant correlations (negative sign) were observed between the discharge at the beginning of the event ($Q_b$), and time from the previous runoff event ($\Delta t$) and the hysteresis magnitude parameter for $NO_3$ ($r = -0.22$, $p < 0.05$). The parameter describing information on the hysteresis direction for $NO_3$ ($\Delta R_{NO3}$) showed negative correlations with rainfall amount (P), maximum 10-min rainfall intensity (IP10), rainfall kinetic energy (KE) and peak discharge ($Q_{max}$). On the contrary, a positive relationship was found between hysteresis direction for TKN ($\Delta R_{TKN}$) and rainfall amount (P), rainfall kinetic energy (KE), peak discharge ($Q_{max}$), total runoff volume (TR) and runoff duration ($S_D$).

**Table 2.** Pearson correlation coefficients between hysteresis descriptors ($\Delta R$ and $\Delta C$) and event characteristics. Values displayed in bold indicate correlation is significant at 0.01 level and italics indicate correlation is significant at 0.05 level.

| | Hysteresis direction ($\Delta R$) | | Hysteresis magnitude ($\Delta C$) | |
|---|---|---|---|---|
| | $NO_3^-$ | TKN | $NO_3^-$ | TKN |
| *Antecedent conditions* | | | | |
| AP7d | -0.18 | 0.09 | -0.19 | -0.08 |
| AP15d | -0.19 | 0.16 | -0.19 | -0.01 |
| $Q_b$ | -0.14 | 0.12 | *-0.22* | 0.02 |
| $\Delta t$ | 0.08 | 0.03 | **0.27** | 0.05 |
| *Event characteristics* | | | | |
| P | *-0.22* | **0.36** | **0.27** | **0.27** |
| IP10 | *-0.24* | 0.00 | 0.05 | *0.25* |
| KE | *-0.24* | **0.32** | *0.24* | **0.31** |
| $Q_{max}$ | **-0.29** | **0.29** | -0.03 | **0.28** |
| TR | -0.17 | **0.38** | -0.08 | 0.04 |
| $\Delta Q$ | -0.06 | 0.18 | **0.29** | **0.35** |
| RL | 0.03 | -0.13 | 0.12 | *-0.23* |
| K | 0.10 | 0.17 | -0.03 | **-0.31** |
| $S_D$ | -0.04 | **0.37** | 0.12 | -0.09 |
| *Concentrations during the event* | | | | |
| $C_{initial}$ | -0.06 | 0.04 | **0.40** | **-0.54** |
| $C_{max}$ | 0.08 | 0.15 | 0.16 | *0.25* |
| $C_{mean}$ | -0.14 | 0.08 | 0.16 | *0.24* |

AP7d: accumulated rainfall 7 days before the event; AP15d: accumulated rainfall 15 days before the event; $Q_b$: discharge at the beginning of the event; P: rainfall amount; IP10: maximum 10-min rainfall intensity; KE: rainfall kinetic energy; $Q_{max}$: peak discharge; TR: total runoff of the event; $\Delta Q$: magnitude of the event relative of the initial baseflow; RL: relative length

of the rising limb; K: slope of the initial phase of the hydrograph falling limb; $S_D$ runoff event duration, $\Delta t$: time from the previous runoff event; $C_{initial:}$ initial concentration; $C_{max}$: maximum concentration; $C_{mean}$: mean concentration.

Regarding the parameters describing the concentration status of $NO_3^-$ ($\Delta C_{NO3}$) and TKN ($\Delta C_{TKN}$), a positive correlation was found among these parameters ($\Delta C_{NO3}$, $\Delta C_{TKN}$) and the hydro-meteorological variables rainfall amount (P), rainfall kinetic energy (KE) and magnitude of the event relative to the initial baseflow ($\Delta Q$). An inverse relationship was found between $\Delta C_{TKN}$ and the relative length of the rising limb (RL) (r = - 0.23, p < 0.01) and slope of the initial phase of the hydrograph falling limb (K) (r = - 0.31, p < 0.01). Finally, the concentrations during runoff events were not controlling factors for the

direction of the hysteresis of $NO_3$ and TKN, but these variables (especially $C_{initial}$) controlled the hysteresis magnitude for $NO_3^-$ ($\Delta C_{NO3}$) and TKN ($\Delta C_{TKN}$), although in different ways (Table 2). Thus, $C_{initial}$ showed positive correlations with $\Delta C_{NO3}$ and negative with $\Delta C_{TKN}$.

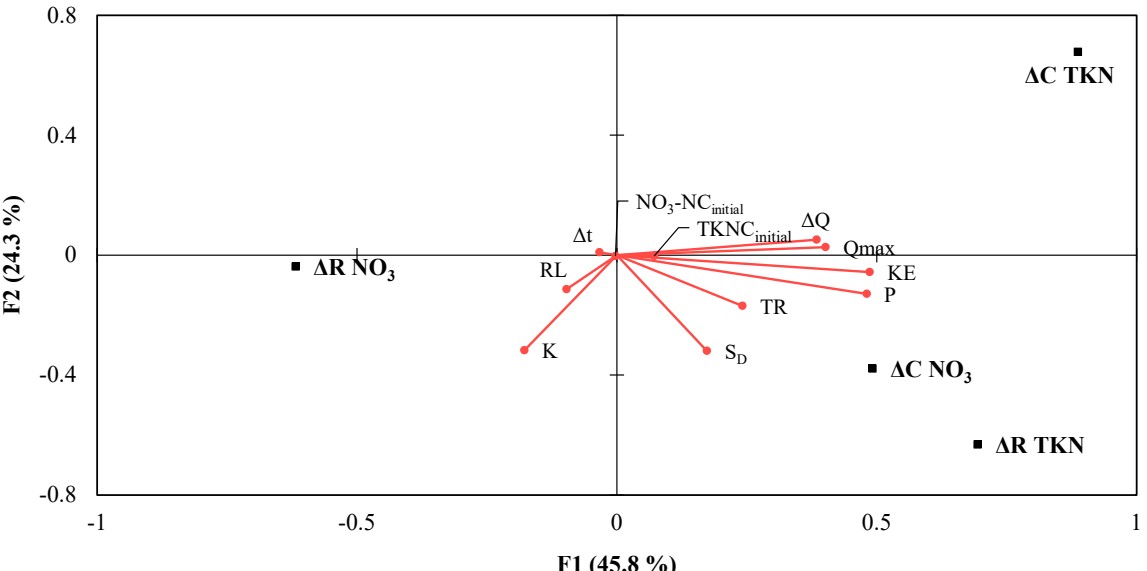

**Figure 7:** Redundancy analysis distance biplot showing ordinations of explanatory and response variables. The response
variables (hysteresis descriptors- $\Delta C_{NO3}$, $\Delta R_{NO3}$, $\Delta C_{TKN}$, $\Delta R_{TKN}$) are shown as black squares and the explanatory variables (i.e. characteristics of the events) as red circles. The red arrows are the vectors of the explanatory variables; the longer they are, the greater their influence. The first two axes of the RDA represent 45.8 % (F1) and 24.3 % (F2) of the explained variance. The angles between the response variables (hysteresis descriptors) and explanatory variables (event characteristic) reflect their correlation, the smaller the angle, the stronger the correlation. P: rainfall amount; KE: rainfall kinetic energy; Qmax: peak discharge;
TR: total runoff of the event; $\Delta Q$: magnitude of the event relative to the initial baseflow; RL: relative length of the rising limb; K: slope of the initial phase of the hydrograph falling limb; $S_D$: runoff event duration, $\Delta t$: time from the previous runoff event; $NO_3$-N or TKN $C_{initial:}$ initial concentration.

RDA analysis confirmed the Pearson´s correlation results. The first two axes explained 70.1% of total variance in the descriptors in $NO_3^-$ and TKN hysteresis ($\Delta C_{NO3}$, $\Delta C_{TKN}$, $\Delta R_{NO3}$, $\Delta R_{TKN}$), accounting for the first and second canonical axes for 45.8 % and 24.3%, respectively. $\Delta C_{TKN}$ and $\Delta R_{TKN}$ loaded positively in the first axis and pointed in the same direction as rainfall-runoff magnitude variables, i.e. rainfall amount (P), rainfall kinetic energy (KE), magnitude of the event relative to the initial baseflow ($\Delta Q$), peak discharge ($Q_{max}$) and total runoff volume (TR), indicating a positive relationship with these. $\Delta R_{TKN}$ and $\Delta C_{NO3}$ loaded negatively in the second axis and pointed in an opposite direction to the slope of the initial phase of the falling limb (K), suggesting that an inverse relationship exists between both variables. On the other hand, $\Delta R_{NO3}$ loaded negatively in the first axis and pointed in the opposite direction to rainfall amount (P), rainfall kinetic energy (KE), magnitude of the event relative to the initial baseflow ($\Delta Q$) and peak discharge ($Q_{max}$), indicating that an inverse relationship occurs with these variables.

**4 Discussion**

High-frequency water quality monitoring facilitates identification of C-Q dynamics at event scale (Lloyd et al., 2016; Vaughan et al., 2017; Rose et al., 2018; Musolff et al., 2021; Winter et al., 2021). We observed that, in general, the $NO_3$-N and TKN concentrations increased during runoff events in comparison to pre-event conditions, indicating the predominance of an enrichment response during runoff events and suggesting that the N delivery in the catchment is mainly controlled by diffuse sources. Nitrate concentrations in drinking water in Europe are restricted to 50 mg L$^{-1}$ as $NO_3^-$ or 11.3 mg L$^{-1}$ $NO_3^-$-N (Directive 98/83/EC), and this limit was not exceeded for the Corbeira catchment. However, considering that in well-oxygenated surface waters, nitrate levels above 0.5-1.0 mg L$^{-1}$ (as $NO_3^-$) can pose a risk of water eutrophication (Camargo and Alonso, 2007) and that 2 mg L$^{-1}$ (as $NO_3^-$) is the threshold identified in the European Nitrogen Assessment as an appropriate target for establishing a river system in good ecological conditions, the data obtained (Table 1) indicate that the study area may be threatened by a potential risk of eutrophication due to increased nitrogen concentration during individual events. It is also important to consider organic nitrogen (expressed as TKN), which accounts for a moderate fraction (>25%) of the total annual N export from the catchment (Rodríguez-Blanco et al., 2015) and can be a source of bioavailable nitrogen through the ammonification and subsequent nitrification of $NH_4$. All of this will clearly have important implications for compliance with water quality targets, and it must be borne in mind that the study area flows into the Abegondo-Cecebre reservoir, a very important source of drinking water for one of the largest cities in the northwest Iberian Peninsula.

4.1 Nitrate and Kjeldahl nitrogen hysteresis patterns

The $NO_3^-$ dynamic during the runoff events was dominated by anticlockwise hysteresis with enrichment. This pattern, in which higher concentrations are observed in the falling limb than the rising limb (Figure 5a, b) is often the result of a greater subsurface flow contribution, followed by that of groundwater (Dupas et al., 2016; Rose et al., 2018; Musolff et al., 2021). Groundwater, which dominates baseflow, contains low $NO_3^-$ concentrations in this catchment. Thus, Rodríguez-Blanco et al.

(2015), showed that NO$_3$-N concentrations in summer (low-flow conditions dominated by groundwater) were lower (around 0.85 mg L$^{-1}$) than those measured in winter (1.28 mg L$^{-1}$). On the other hand, the soils in the study area have high infiltration rates (Taboada-Castro et al., 1999) and a great deal of streamflow comprises water leached through the soil profile (Rodríguez-Blanco et al., 2019), so subsurface flow is a likely pathway delivering additional NO$_3^-$ during runoff events. Previous studies carried out in the catchment to make an approximate estimation of the contribution of surface flow and subsurface flow to direct runoff, using electrical conductivity as a tracer, have shown that subsurface flow constitutes the main route of generating direct runoff in the catchment (mean value 72%, 39-90%; Rodríguez-Blanco, 2009). In addition, the largest contribution by the subsurface flow has been documented at the beginning of the falling limb (Rodríguez-Blanco, 2009), that is, when the highest NO$_3$ concentrations were recorded. This pattern (anticlockwise) could also be related to the higher availability of NO$_3^-$ in the shallow soil horizon than in deeper groundwater, arguably because of mineralization of organic matter and the mineral and organic fertilizer applied to agricultural soils in the catchment, as demonstrated by both modelling and data-driven approaches (López Periago et al., 2002). Other authors, however, have attributed anticlockwise hysteresis to a particular spatial distribution of the sources. Thus, Vaughan et al. (2017), in a study in a heavily fertilized agricultural catchment, linked this pattern to the transport of proximal sources at the beginning of runoff events, followed by enrichment from distal and substantial sources of nitrate, which seems not to be the case in the study catchment, because this pattern, i.e. anticlockwise with enrichment, was also observed in low volume runoff events, so it does not seem feasible that distal sources of NO$_3^-$ can be activated in these events. However, in large runoff events the areas of the catchment acting as nitrate sources could increase as hydrologic pathways connect and NO$_3^-$ also arrives from distant sources. While enrichment with anticlockwise rotation was the dominant pattern for NO$_3^-$ in the study catchment, some events exhibited enrichment patterns but with clockwise hysteresis (Figure 5 and 6), indicating a higher NO$_3^-$ concentration in the rising limb of the hydrograph than in the falling limb. This pattern was mainly observed in small spring events occurring in 2008 and 2009 and could be attributed to the transport of NO$_3^-$ from near-stream areas that have received fertilizers.

The dominance of anticlockwise nitrate hysteresis with higher concentrations on the falling limb of the hydrograph is in line with findings from reported studies conducted in forest and agricultural catchments (Butturini et al., 2006; Cerro et al., 2014; Outram et al., 2016; Musolff et al., 2021). However, they contrast with interpretations presented in several previous studies carried out in rural catchments, which only described NO$_3^-$ dilution processes linked to dilution of NO$_3^-$ concentrations in groundwater by surface runoff (Bowes et al., 2015; Aguilera and Melack, 2018; Rose et al., 2018; D'Amario et al., 2021) or the exhaustion of a finite pool of NO$_3^-$ in the riparian zone and shallow groundwater (Koenig et al., 2017; Duncan et al., 2017). However, in the study catchment, the dilution responses (in an anticlockwise direction ($\Delta C \leq 0\%$)) were only observed in certain larger runoff events associated with wet antecedent conditions and a short interarrival time ($\Delta t < 1$ hour). A possible reason for the initial dilution of the concentrations could be the preceding wetting of the catchment which favours the delivery of relatively low-nitrate water flushed to the stream from low- NO$_3^-$ concentration sources, such as direct rainfall, i.e. rainfall (mean NO$_3^-$ concentrations determined in rainwater samples = 0.1 mg NO$_3^-$ L$^{-1}$, Rodríguez-Blanco et al., 2015) falling directly into the stream or near saturated areas around the stream and runoff from roads and paved area, as has been observed in other

headwater streams (Poor and McDonnell 2007; Kato et al., 2009). Following the initial dilution, concentrations increased above pre-event values, reaching the highest $NO_3^-$ concentrations after maximum discharge. The return of $NO_3^-$ concentrations to the values before the rainfall-runoff event is especially slow in these cases (Figure 5b) and $NO_3^-$ concentrations remain elevated for several days until streamflow returns to baseflow.

Mechanisms responsible for TKN mobilization differ from those mobilizing $NO_3^-$. Thus, a clockwise enrichment pattern for TKN concentration was dominant for most events (Figure 5d), suggesting a delivery of TKN to the stream network via fast pathways from proximal sources or relatively easily connected to the stream as event discharge increases, with possible rapid exhaustion of the material to be transported (Creed et al., 2015). The TKN response was almost concurrent with discharge, which leads to thinking that TKN may come from the eroded soil and litter layer delivered to the stream primarily by surface runoff, in a similar way to suspended sediment matter and particulate phosphorus (Rodríguez-Blanco et al., 2013; 2019). These results generally agree with many that also reported clockwise organic nitrogen hysteresis patterns (Vanderbilt et al., 2003; Inamdar and Mitchell, 2006; Rose et al., 2018; D'Amario et al., 2021). In contrast, Hagedorn et al. (2000) observed anticlockwise hysteresis for organic nitrogen in a forest catchment in Switzerland due to its passage through the forest canopy and organic-rich topsoil. Others have attributed this pattern (anticlockwise) to distant source areas activated later in the runoff events as hydrological pathways connect (Aguilera and Melack, 2018). In the study area, this pattern (anticlockwise) was only observed during some low-volume runoff events caused by low rainfall events (specifically, 20 of the 26 events with this anticlockwise pattern were associated with less than 16 mm of rainfall) recorded in spring and summer, so the delivery from distant sources is less likely to explain the anticlockwise pattern due to the reduced hydrological response of these events. Rather, given the event characteristics (low-volume rainfall and runoff events with little particulate material), the presence of anticlockwise hysteresis could be indicative of a large proportion of dissolved fraction in TKN, which passes through the soil and subsequently enters the stream by subsurface flow.

Hysteresis patterns may vary among events and antecedent soil moisture conditions are often recognized as an important factor in the response of different constituent concentrations among events, even when rainfall characteristics are approximately similar (Butturini et al., 2006; Baker and Showers, 2019). However, other authors have highlighted the important role played by rainfall-runoff events characteristic on hysteresis patterns (Chen et al., 2012; Lloyd et al., 2016). For example, Chen et al. (2012) emphasized the role of the runoff event magnitude influencing the magnitude and direction of the hysteresis patterns, whereas Lloyd et al., (2016) underlined the combined effect exerted by storm duration, maximum discharge during the runoff event and the time elapsing from the previous runoff events on controlling N hysteresis magnitude and rotation. In this catchment, hysteresis direction and magnitude for TKN were better explained by event characteristics, such as rainfall amount (P), peak discharge ($Q_{max}$), and event magnitude relative to the initial baseflow ($\Delta Q$)) than by antecedent rainfall (AP7d, AP15d) and baseflow ($Q_b$). Thus, the hysteresis magnitude value was dependent on the magnitude of the hydrological response of the catchment and the delivery of particulate material to the stream. In the catchment, the main sediment supply to the stream is associated with the erosion of cultivated soil with high connectivity to the stream, which favors the quick delivery of particulate material to the stream. The results obtained for TKN are consistent with the findings obtained for suspended

sediments, phosphorus, and particulate metals in the study area by Rodríguez-Blanco et al. (2010, 2013, 2018), showing that sources of particulate material are close to the monitoring station, which may explain the prevalence of clockwise hysteresis for TKN. For $NO_3^-$ hysteresis patterns the role of hydrometeorological conditions were more complex and dynamics should be controlled by biogeochemical processes coupled with hydrological processes Butturini et al., 2006; Heathwait e and Bieroza ,2020). For example, Heathwaite and Bieroza (2020) when analysing the interplay between hydrological flushing and biogeochemical cycling during high and low-magnitude events in an agricultural catchment in the UK, emphasized that the pattern of $NO_3^-$ mobilization is controlled by the magnitude of the runoff event, with high-magnitude runoff events driving rapid mobilization (clockwise hysteresis with dilution) and low-magnitude runoff events driving delayed mobilization (anticlockwise hysteresis with enrichment). In the study catchment, the $NO_3^-$ hysteresis magnitude ($\Delta C$) was related to the initial $NO_3$ concentrations ($NO_3$-$NC_{initial}$), magnitude of the event relative to the initial baseflow $\Delta Q$) and the time elapsed since a preceding runoff event ($\Delta t$), which highlights that, beyond the magnitude of the runoff event-essential driver for $NO_3^-$ hysteresis magnitude variability, the inter-event period, during which physical and biological processes operate to increase the store of available nutrients and sediments (Walling and Webb, 1982), as well as the pre-event biogeochemical conditions, also influenced $NO_3^-$ hysteresis magnitude ($\Delta C$). However, this last driver does not appear as the most relevant, indicating that initial $NO_3$ concentrations ($NO_3$-$NC_{initial}$) play a more subtle and diffuse role (Figure 7).

**5 Conclusion**

Our study uses measurements of stream discharge and nitrogen ($NO_3^-$ and TKN) obtained by intensive sampling to investigate the nitrogen concentration dynamics at event scale. The results show the potential of high-frequency N concentration monitoring to advance our understanding of coupled hydrological and biogeochemical systems in the context of contrasting hydrometeorological conditions. Assessment of nitrogen C-Q relationships and their controlling factors has provided evidence of the different $NO_3^-$ and TKN dynamics during the runoff events, suggesting the presence of distinct delivery mechanisms and differences in dominant hydrological pathways. $NO_3^-$ behaviour during the runoff events was dominated by anticlockwise hysteresis, whereas clockwise hysteresis prevailed in the TKN dynamic. However, for both solutes ($NO_3^-$ and TKN), the magnitude of the hydrological response played a primary role in controlling the magnitude of hysteresis variability.

The divergence dynamics observed between N components in the study area highlights the need to understand the transportation of N and the mechanism for controlling the implementation of future water quality monitoring programs and the development of N-specific management plans to ensure that control measures are most effective at the catchment scale, especially within the context of increasing nitrate concentrations, a pressing environment issue. Thus, to minimize nitrate delivery to streams, catchment management should focus on reducing N stores in the soil, whereas for protecting water quality against TKN, management options to reduce surface runoff and sediment are also required, since they seem to be mainly responsible for TKN transport in this region.

*Data availability*. The data that support the findings of this study are available from the corresponding author, PET, upon
reasonable request.

*Author contributions*. The original draft of the manuscript was prepared by MLRB, and reviewing and editing were provided
by MTTC and MMTC. F. All authors have read and agreed to the current version of the paper.

*Competing interests*. The authors declare that they have no conflict of interest.

*Acknowledgements*. This research was carried out within the projects REN2003-08143, funded by the Spanish Ministry of
Education and Science, and PGIDIT05RAG10303PR and 10MDS103031PR, financed by the Xunta of Galicia. We would like
to thank two anonymous reviewers for their helpful comments and suggestions.

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
