# Peer review of "Improving the understanding of N transport in a rural catchment under Atlantic climate conditions from analysis of the concentrationdischarge relationship derived from a high frequency data set"

_Hydrology and Earth System Sciences, 2021_

## Author Response (AR1)

Dear Reviewer,

Thank you for the time you devoted to reading this manuscript and for your helpful comments. We took all your remarks and suggestions into account to improve our manuscript

**General comments:**

1. The novelty of the study is not sufficiently highlighted. The additional value of the performed analysis is not clear because the Introduction does not provide a sufficient overview of findings on the controls of nitrate hysteresis from the existing high-frequency studies. Particularly, the Introduction should clearly indicate what is known so far about the role of event characteristics on the hysteresis patterns and what is the additional value of the analysis performed in this study. Moreover, it should be indicated what kind of additional information the analysis of TKN might provide compared to the analysis of nitrate concentrations alone.

Following your advice, we have rewritten the introduction. See L. 56-80.

2.- There are several methodological ambiguities and subjective choices that have to be clarified, particularly concerning the necessity and additional value of redundancy analysis (see detailed comments to Lines 206-210), separation of event and pre-event water that was performed but was not further analyzed (see detailed comments to Lines 159-168), the choice of subjective manual techniques for event identification and hysteresis classification (see detailed comments to Lines 135-139 and Lines 174, respectively).

Following your advice, the section selection of runoff events and description of C-Q hysteresis was clarified. See L. 150-154

1. The rationale for the analysis of two selected substances should be clearly stated, as well as the rationale for the choice of event characteristics. Why are particularly these event characteristics expected to be decisive for hysteresis patterns? A more hypothesis-oriented choice of these characteristics might also be helpful to highlight the novelty and the additional value of this study compared to the previous literature.

Following your advice, we have clarified this in the text. See L. 155-156.

1. The connections of different hysteresis patterns to the particular runoff generation processes in the Discussion Section are rather speculative and need more in-depth clarifications and evidence from related field studies or own observations.

Following your indications, we have improved the discussion section. See L. 329-405

Clarifications

1. There is little discussion provided on the identified relations between event characteristics and nitrate and TKN hysteresis patterns in the Discussion Section, although it appears to be the central topic of the manuscript.

This information was added in revised manuscript. See L. 390-405.

Detailed comments

Line 12: The title implies the analysis of several catchments, although in fact the analysis was performed in a single catchment. Please revise.

Our apologies, we have corrected this.

Line 17: Not clear what is meant here by the overall dynamics of hysteresis. Please clarify.

We have deleted overall

Line 18: Not clear what is meant here by "parameters". Please clarify.

Our apologies, we have modified this.

Line 19: Please be more cautious with such statements. It is rather difficult to infer particular runoff generation processes directly from hysteresis patterns. Please revise.

It was revised in the new version of the manuscript.

Lines 19-21: Consider providing here an explanation why there are such considerable differences in the hysteresis patterns of NO3 and TKN.

The explanation was included in the discussion section.

Line 22: Consider providing more details on how exactly different event characteristics affect corresponding hysteresis patterns. Moreover, please clarify what is the difference between "runoff" and "discharge" here.

This information was added. See L. 20-23.

Lines 29-31:    This is rather general, the Introduction can benefit from a more specific opening sentence.

The sentence was replaced. See L. 30.

Lines 46-47, 50-53, 53-54, 56-57, 62-64:    These statements require a reference.

The references were added. See L. 40-80

Lines 59-60:    I cannot agree with this statement. The analysis of nitrate hysteresis are rather often performed in the headwater catchments since high frequency observations are usually only available from local research observatories (see e.g., Knapp et al.,

2020; Winter et al., 2020; Mussolf et al, 2021; Koenig et al., 2010; Vaughan et al., 2017 among many others).

Following your advice, we have modified this paragraph.

Line 62: What is meant here by "clean" rural catchment? Please clarify.

Our apologies, the paragraph have been replaced.

Lines 71-72: What these drivers can be? This part of the Introduction should provide a clear rationale on selecting event characteristics for the analysis based on findings from previous studies and/or own hypothesis on which characteristics might be potentially important for the hysteresis of NO3 and TKN concentrations.

The objectives were rewritten. See L. 83-88.

Lines 72-74: Are such studies becoming increasingly rare? I would argue that CQ studies become increasingly frequent as the density of observations has increased in the past decades. Please clarify.

Our apologies, the paragraph was rewritten following your comments. 55-60

Line 74: Please clarify why there is a particular interest in the concentrations of TKN and how its analysis complements NO3 investigations.

This information was added. See L. 72-80

Lines 69-85: The structure of the Introduction and especially of this last paragraph is rather confusing. The Introduction has to be streamlined and clearly present the current state of the art on the topic of nitrate hysteresis and its potential controls, indicate the knowledge gap and provide clear objectives of this study that strive to close this gap.

The introduction and objectives were rewritten.

Line 90: What is meant here by relief? The difference between max and min elevation? Please clarify.

It was clarified. See L. 101.

Figure 1: Please indicate river outlet and river network on this Figure. Please clarify the title of the legend. What is shown by the solid line within the catchment? Please clarify.

The information was added. Solid line is the rail network. See Figure 1.

Line 106: This number of meteorological stations does not seem very plausible to me. Please correct and consider displaying the locations of the considered meteorological stations in Figure 1.

The ID of the meteorological station is corrected; 10045

Line 113: Please clarify which N concentrations exactly were measured.

The N concentrations measured were added. See L. 125

Lines 112-115: Please indicate how many gauges were actually used for interpolation, consider indicating their location in Figure 1.

Three rainfall gauges were used, it was indicated in the manuscript. See L. 125.

Lines 121-122: Please clarify how the start of the rainfall event was identified.

It was clarified. See L. 149-150.

Lines 123-124: Please clarify how exactly sampling frequency was defined based on magnitude and duration of the event? Were the forecasted values used? If so, please indicate that and provide details on the type and the accuracy of the forecast.

Information was added. See L. 135-136.

Lines 135-137: Does this threshold correspond to the rule described in the Line 121-122? Moreover, please indicate if any minimal interarrival time between rainfall events was used for event definition.

Information was added. See L. 150

Lines 137-138: Please clarify how the inflection point is identified.

Lines 138-139: Please clarify how baseflow conditions were identified. Was baseflow separated? Which method was used for that? How is the start of the next event defined?

This information was added. See L. 150-154

Lines 142-148: The choice of these particular event characteristics has to be clarified. Why exactly these characteristics? What do authors expect to find by investigating them? Moreover, some of the characteristics e.g., water yield, rainfall kinetic energy have to be introduced in more detail, i.e., source, definition etc.

This information was added in the revised manuscript. See L. 155-166

Line 145: This definition of delta Q will not result in % units. Please revise.

It was revise. See L. 163. $\Delta Q$; i.e., $(Q_{max}-Q_b)/Q_b*100$, %)

Line 146: Why the duration of the events is in days when the observations are performed every 10 min? Please clarify.

The duration of the events now is provided in h.

Line 147: How was the initial phase of the falling limb identified? Please clarify.

We are referring to slope of the falling limb; i.e. after peak discharge. See L. 165

Line 151: Please clarify what is meant here by "metal load".

It is a mistake; it is referring to N load

Our apologies, it is a mistake. It was corrected. See L. 165.

Lines 159-168: Please notice that EC is not always applicable for pre-event and event water separation (see e.g., Musolff et al. 2015; Musolff et al., 2020). Please indicate if implemented assumptions are expected to be valid in the study area. Moreover, the results of this separation were barely mentioned in the Results section and are absent from the Discussion altogether. Consider either removing this separation or including it more distinctly in the analysis of hysteresis patterns. Generally, it is not clear what additional insights about the controls of hysteresis patterns can be gained from this analysis.

Following the indications of two reviewers the hydrograph separation using EC was deleted.

Lines 171-172: Please clarify why multiple-peak events cannot be considered.

In multiple-peak events the relationship between N concentration and discharge is difficult to define (e.g. several peaks of N concentrations)

Lines 173-175: Please clarify why visual examination was preferred to the automated approaches. How can the robustness of the performed classification be verified?

The visual examination was used for verification

Line 177: Please indicate what is exactly meant by "figure of eight".

We refer to figure-eight hysteresis, which were classified as clockwise or anticlockwise depending on the succession of the peak concentration and peak discharge, in a similar way to Bieroza and Heathwaite (2015). See L. 181-183.

Table 1: Please explain what is V.C.? Consider including delta_t to the "antecedent conditions" group, as it rather represents pre-event than event conditions. Moreover, please clarify why it has units of days when the observations are available on much finer resolution? Is that the reason for its min value being equal to zero? This is rather confusing as it makes an impression that a consecutive event can start at the same time step when the previous event finishes.

Following your comments, Table 1 was improved.

CV: coefficient of variation.

delta_t was included into antecedent condition. The units were changed to hours. In same cases, consecutive runoff events can occur.

Line 192: Please clarify what AR stands for? Generally, the manuscript is oversaturated with many not very intuitive acronyms. Consider using full terms instead.

The information was added. See L. 189.

Line 195: Please clarify how standardization is performed here. Was any standardization applied for Ah?

Yes, it was clarified in the revised manuscript. See L. 202-207.

Lines 200-204: This part is rather confusing and hard to understand, consider revising. Moreover, please indicate if the selected thresholds are in line with previous hysteresis classifications in the literature.

Following your indications, the paragraph was rewritten. See L. 214-218.

Lines 209-210, 285: From this description, it is not clear what is the additional value of the redundancy analysis compared to correlation. Please clarify.

Additional information was provided in the revised manuscript. See L. 223-225.

Figure 2: Please clearly state in the caption what these four panels show (i.e., time of event, type of hysteresis). Please add a-d labels to the panels. Please clarify why four different hysteresis are displayed when only three types were considered? Do the two plots on the top correspond to the same type or not? Please indicate the starting point of the event in each subplot.

Figure 2 (5 in the revised manuscript) was modified, and the caption was improved. See Figure 5

Lines 217-219: It is not clear how the authors are able to define that the entire range of rainfall and antecedent rainfall is covered without examining what their actual range is. Please clarify.

The sentence was rewritten. See L. 234-236.

Lines 224-226, 232-233: Seasons are not indicated on Figure 2 making it impossible to verify statements in these sentences. Please add corresponding information to Figure 2 or revise these sentences.

Figure 5 (before 2) indicates the time of the runoff events.

Line 227: Please indicate what is considered here by "long duration"

It refers to runoff events lasting several days.

Line 244 and Figure 3: Do these type numbers correspond to Figure 3? Please indicate this in the text and in the caption of Figure 3.

The information was added in the revised manuscript. See L. 264, 271.

Line 246: In case of NO3, only 62% of events have positive delta C. This is not very similar to 93% for TKN. Please revise.

Our apologies, it was deleted.

Line 246: Compared to baseflow or to the pre-event values? Baseflow was not formally separated (at least the Method section provides no indication of such analysis). Please revise.

It was revised.

Lines 296: Please clarify what kind of information is provided in the parenthesis.

It refers to N-NO3 concentrations. See L. 320.

Lines 305-307: Anticlockwise hysteresis was also linked previously to a particular spatial distribution of sources (see e.g., Vaughan et al., 2017). Please indicate if this can also be the case in this study catchment.

This seems not be case in the study catchment. See L. 341-344

Lines 312-315: Please indicate how the point with maximum contribution of subsurface flow can be identified here.

It was identified from EC. See L. 334-335.

Line 315-316    : Such a statement requires references. Please add.

The reference was added. See L. 341.

Lines 319-321: Please clarify why particularly in these years such conditions have arisen. Moreover, this sentence is rather confusing, please revise.

The sentence was revised.

Lines 326-328: Is there any evidence of surface runoff presence in this particular study catchment? Please clarify.

Although surface runoff only represents a small percentage of the flow, we have field evidence of surface runoff in particular events.

Lines 329-330: This seems like a description of the "eight" hysteresis shape that was not considered in the classification in this paper. Please revise,

No, it is not a description of figure-eight

Lines 331: Please indicate clearly which event from Figure 2 is meant here.

The information was added. See L. 363.

Lines 332-333: It is not clear how this confirms the control by subsurface flow. Please be cautious with your conclusions if they cannot be directly supported by your own findings. Please revise.

It was deleted in the revised manuscript.

Line 336: Please clarify what kind of findings or observations support rapid exhaustion in this case?

The reduction of particulate matter.

Lines 349-351: This sentence is rather confusing. Which event characteristics point this out? What can be a possible source of this additional nitrogen? Please clarify and revise.

The sentence was rewritten in the revised manuscript. See L. 378-380.

Lines 355: What is the difference here between runoff and discharge. Please clarify.

The sentence was modified in the revised manuscript. See L. 388-395.

Line 354-356: The dominant role of event characteristics, particularly of rainfall intensity and by extension its connection with the activation of fast flow paths, stated in these sentences in my opinion contradicts the previous statement in Line 332-333 about the dominant control of subsurface flow. Please clarify.

This was modified in the revised manuscript. See L. 388-395.

Lines 356-357: The relation between rainfall intensity, discharge rates and nitrogen loss is not clear from this description. Please provide more process-oriented hypotheses on how event characteristics might affect hysteresis.

It was rewritten in the revised manuscript. See L. 395-398.

Line 359-361: This rather contradicts earlier statements (Line 232-233) that nitrate concentration in winter (wet season) is higher than in any other season. Please clarify.

It was deleted in the revised manuscript.

Line 363: What is meant here by "strength of the event". Please clarify.

It was deleted in the revised manuscript.

Line 368: What do you mean here by "losses" here? Consider using a more conventional term here.

The term losses was substituted by delivery. See L. 401

Lines 368-370: This statement would be much more clear if the main sources of TKN were introduced earlier. This is the first time they are mentioned in the manuscript. Please revise.

It was revised.

Lines 377-379: Be cautious providing statements that are not directly inferable from your own results. It is rather difficult to identify dominant runoff generation processes from hysteresis patterns alone.

Following your comments, it was deleted in the revised manuscript.

**Editorial comments**

Lines 14,19,69 and elsewhere: Consider using the correct chemical "NO3-" notation.

Following your indications, we have used NO3- in the revised manuscript

Line 150: Consider using "initial" instead of "0".

Following your comment, we have used Cinitial.

Line 182: Consider using hysteresis "characteristics" instead of "parameters".

Lines 183, 190: The term "trend" is not clear here. Consider using the term "slope" instead.

Our apologies, we have substituted the term trend by slope

Line 185: Please add "ΔC=" on the left side of equation 2

Line 199, 203: consider using "classes" instead of "regions" as these cases do not have any spatial aspect.

Ok, the term regions were replaced.

Line 229: word order: "with discharge" should be before "were observed"

Thank you for the correction. See L. 248.

Line 238: than before the event.

Ok, it was corrected.

Line 239: an increase in NO3

Thank you for the correction. See L. 258.

Line 253: repetition "10%"

Our apologies, it was corrected. See L. 271.

Table 2: In the caption "bold" instead of "both". Consider transforming this table into a correlation matrix to improve visualization of the results.

Thank you for the correction.

Line 301: due to increased nitrogen concentrations

Thank you for the correction. See L. 325

Lien 304: they are

The sentence was deleted in the revised manuscript.

Line 305: it should be Winter et al. 2021

Thank you very much for the correction.

Line 314: contribution of subsurface flow

Line 325: runoff events with short interarrival time

Line 349: distant sources

Thank you for the correction. See L. 375.

Line 359: under wet antecedent conditions?

Thank you for the correction, but the sentence was deleted in the revised manuscript

Line 372: Conclusion or Concluding remarks

Ours apologies; the name of the section was corrected. See L.406.

Line 375: C-Q

Thank you for the correction.

**References**

Knapp, J. L. A., von Freyberg, J., Studer, B., Kiewiet, L., & Kirchner, J. W. (2020). Concentration–discharge relationships vary among hydrological events, reflecting differences in event characteristics. *Hydrology and Earth System Sciences*, *24*(5), 2561–2576. https://doi.org/10.5194/hess-24-2561-2020

Koenig, L. E., Shattuck, M. D., Snyder, L. E., Potter, J. D., & McDowell, W. H. (2017). Deconstructing the Effects of Flow on DOC, Nitrate, and Major Ion Interactions Using a High-Frequency Aquatic Sensor Network. *Water Resources Research*, *53*(12), 10655–10673. https://doi.org/10.1002/2017WR020739

Musolff, A., Schmidt, C., Selle, B., & Fleckenstein, J. H. (2015). Catchment controls on solute export. *Advances in Water Resources*, *86*, 133–146. https://doi.org/10.1016/j.advwatres.2015.09.026

Musolff, A., Zhan, Q., Dupas, R., Minaudo, C., Fleckenstein, J., Rode, M., Dehaspe, J., & Rinke, K. (2021). Spatial and Temporal Variability in Concentration□□Discharge Relationships at the Event Scale. *Water Resources Research*, *57*. https://doi.org/10.1029/2020WR029442

Vaughan, M. C. H., Bowden, W. B., Shanley, J. B., Vermilyea, A., Sleeper, R., Gold, A. J., Pradhanang, S. M., Inamdar, S. P., Levia, D. F., Andres, A. S., Birgand, F., & Schroth, A. W. (2017). High-frequency dissolved organic carbon and nitrate measurements reveal differences in storm hysteresis and loading in relation to land cover and seasonality. *Water Resources Research*, *53*(7), 5345–5363. https://doi.org/10.1002/2017WR020491

Winter, C., Lutz, S. R., Musolff, A., Kumar, R., Weber, M., & Fleckenstein, J. H. (2021). Disentangling the Impact of Catchment Heterogeneity on Nitrate Export Dynamics From Event to Long-Term Time Scales. *Water Resources Research*, *57*(1), e2020WR027992. https://doi.org/10.1029/2020WR027992

Thank you for the references. They were used in the revised manuscript.

Dear Reviewer,

Thank you for the time you devoted to reading this manuscript and for your helpful comments. We took your remarks and suggestions into account to improve our manuscript

Understanding nutrient transport in rural catchments is important for managing water quality and protecting ecosystems. Detailed studies such as this are valuable in understanding the sources, fluxes, and transport mechanisms of nutrients. However, the paper needs revising before it us suitable for publication.

My main general criticism of the paper is that it does not place the results in context. The Introduction (L29-45) presents the global overview and discusses some of the important issues. However, the Section 5 is not very informative (as section 4 is also the Discussion, I presume that this is meant to be the Conclusions). For papers such as this to appeal to a broad international readership, those important issues need to be revisited at the end of the paper and the authors need to explain how their study informs work done elsewhere. Leaving the reader to work that out for themselves is not satisfactory. So, explain here what the implications are and/or how the work has advanced our understanding in general.

 The introduction and discussion sections were widely rewritten.

Some of the methodology is not well explained and it is difficult to follow the details from the figures and tables provided. More details are needed in places (better / additional figures and perhaps some supplementary tables). Moreover, where are the data? Even at the preprint stage, I would have expected the data to have been provided.

The methodology was clarified, and some figures were added in the revised manuscript.

The English is understandable but idiomatic in places and would benefit from a final editing. The paper also is difficult to read in places due to the large number of abbreviations. Mostly, they are introduced, but are easy to lose track of. Most variables need abbreviating, but other abbreviations (eg RDA) probably do not need to be there.

 The English was checked by a native and some abbreviations were replaced by the full name

Abstract

The Abstract provides a good summary of the paper. However, as with the paper as a whole, provide one or two sentences at the end which explain why this is important.

One sentence was added at the end of the abstract. See L. 22-25.

Minor comments

L17-18. "Some metrics" is redundant, meaning of "overall dynamics" is unclear. Perhaps also specify what you mean by "nitrogen behaviour".

Our apologies, these terms were deleted in the revised manuscript.

L18. Sentence "The results showed…" is redundant as this is described in the next sentence

Our apologies, these terms were deleted in the revised manuscript.

L19-20. Does TKN also show dilution, if so mention that here.

This sentence was modified.

L22-23. This explains what you think is important but not how or why. Can you briefly expand on this?

One sentence was added at the end of the abstract. See L. 22-25.

Introduction

As noted above, the first paragraph of the introduction introduces some important general issues that need to be addressed throughout the paper, especially at the end.

The first paragraph was changes following the indications of another reviewer. The importance of findings was added at the end of the discussion section. See L. 399-405.

The referencing in parts of the introduction can be improved. For example, several of the references in L33-45 are reports and Bieroza et al., 2018; D'Amario et al., 2021 deal with techniques. Try to add a few key papers that discuss these general issues.

Some references were added in the introduction. See L. 35-55

L46-47. Not clear what you mean here.

The sentence was modified in the revised manuscript. See L. 41-43

L50-54. Some of the concepts here could also use better referencing. Evans & Davies (1998: Water Resources Research, 34, 129-137) and Walling and Foster (1975: Journal of Hydrology, 26, 237-244) present some of the framework for these studies. Lloyd et al. (2016: Hydrology and Earth Systems Sciences, 20, 625-632) develop a framework for characterising hysteresis and also reference other literature. The recent paper by Knapp et al. (2020: Hydrology and Earth System Sciences, 24, 2561-2576) also develops methodology that is relevant here.

Thank you very much. The references were added. See L. 43-55.

L59-65. This may be generally the case, but there are several studies that have looked at smaller catchments at high frequency. Some examples: Lloyd et al. (2016, Science of the Total Environment, 543, 388-404); Vaughan et al. (2017: Water Resources Research, 53, 5345-5363); Jiang et al. (2010: Soil Sci Plant Nutr, 56, 72-85). Other papers (eg Knapp et al., 2020) apply similar methodology to other solutes (including other nutrients such as P). The way that this is written implies that there is little work being done in this space, when there is a large body of work that needs acknowledging.

Our apologies. This part of the introduction was totally rewritten. See L. 56-82.

L48-58. You could add a schematic figure of hysteresis loops here to show differences in typology. This would make it clear exactly what you mean by rotation directions, slopes etc. You can also highlight what ΔC and ΔR represent.  Alternatively, you could better annotate Fig. 2 to show these features.

Following your recommendations, figures 1 and 4 were added. In addition, the figure 2 (figure 5 in the revised manuscript) was improved. See figures in the revised manuscript.

L70-85. The objectives are very specific and parochial. While they describe what you have done and why it is locally important, can you reframe them so that they have a more general focus (ie understanding differences in the behaviour of TKN and NO3 that we do not understand in general)

The objectives were rewritten in the revised manuscript. See L. 83-88.

L73-74. The studies cannot become increasingly rare – do you mean that there have not been many thus far?

Our apologies, it was a mistake.

L74-76. If understanding TKN is important, you could mention it in the main part of the introduction with a few more details rather than in your objectives section. It is a bit lost here.

 It was mentioned in the revised manuscript. See L. 72-80.

Materials and Methods

The description of the Study Site (Section 2.1) is comprehensive.

L105-108. Is this the meteorology of the site or of the region (not sure if there are 10045 stations, which seems a lot, or if that is the station identifier and the sentence is slightly misworded).

10045 is the ID of the meteorological station. See L. 115

L108. What do you mean by "pluvial" in this context?

It means that hydrological regime follows the rainfall.

L113. How was the hydrological year defined – specify the date / month when it starts.

This information was added in the revised manuscript. See L. 124

Figure 1 only clearly shows landuse. Highlight the drainage features and the monitoring point(s).

The information was added in the revised manuscript. See figure 2 in the revised manuscript

L125-128. The preservation and storage of samples presumably is only after they are retrieved from the autosampler. How much of a delay between collection and storage was there and does this have an impact (I presume not).

The samples were removed from the autosampler within a few hours after runoff events. See L. 137-138.

L125-132. Minor point, but here and elsewhere be consistent with specifying valences or not.

Our apologies, we try to be consistent.

L135-139. For clarity, can you include a figure of a typical event showing rainfall, streamflow, and concentration data. That would help visualise the data and interpretation of events etc.

Following your comment, figure 3 was added in the revised manuscript

L140-151. These is some repetition here as you say that there are three groups of variables, describe broadly what they are, and then explain that in more detail. Some of these variables do not seem to have been explained – how did you calculate KE and I presume that WY is Q / catchment area? Some of the parameters associated with the Q-C loops would be clearer if they were on a figure (previous comment).

The description of the variables was improved. See L. 155-163. In addition, a schematic representation of the hysteresis loops was added. See figure 4 in the revised manuscript.

L145-146. Qmax-Qb will not give you a %

$\Delta Q$ is magnitude of the event relative to the initial baseflow ($\Delta Q$; i.e., (Qmax-Qb)/Qb*100, %). See L. 163.

L151. Metal?

Our apologies, it is a mistake.

L154-159. There are probably better references.  Yu & Schwartz (1999, Hydrol. Process., 13, 191–209) has an early general explanation. For these general statements, try to quote some of the early papers that develop the techniques or review-style papers.

Following the comments of the reviewers, the paragraph was deleted in the revised mansucript.

L154-168. The use of EC in this way makes a range of assumptions (e.g., that pre-rainfall EC represents water from within the catchment, not a mix of that water and recent prior rainfall; that we know the EC of surface runoff; that the contrast in rainfall and catchment water EC is high). These are discussed in many papers that have used that technique (e.g. Miller et al., 2014, Water Resour. Res., 50, 6986–6999; Miller et al., 2016, Water Resour. Res., 52, 330–347; Riis et al., 2015, J. Hydrol. Reg. Stud., 4, 91–107; Rumsey et al., 2017, Hydrol. Process., 31, 4705–4718). Some comments are needed here if you are going to use that technique. Actually, I am not sure whether you really need this parameter – there is a little discussion in section 3.3 but not much else (?). Given the considerable uncertainties in using an EC mass balance, you probably could safely omit it.

Following the comments of the reviewers, the paragraph was deleted in the revised mansucript.

L193 Delta (Δ) R not AR?

Thank you for the correction

Table 1. What is V.C.?

Our apologies. CV is the coefficient of variation. See L. 185

L203 Linear not lineal

Thank you for the correction.

L206-210. This just says that these methods were used. In particular is there any reason that the Redundancy Analysis offers more than the correlations? Any details that we should know?

Some information about RDA was added. See L. 223-225

Results

Figure 2. Without some more details, it is difficult to interpret this diagram. I presume a & b have similar rotation but different slopes? How close to the start of the events are the first points and what time periods do these events depict (do the points within each graph and between graphs represent the same timesteps). More could be done to make it clear what is going on here.

Some information was added to figure 2, i.e., figure 5 in the revised manuscript.

L223-233. There are a lot of generalities here ("highs", "lows" etc) and Table 1 only shows summary statistics. Add a few more details to the text to explain and consider adding a Supplement Table with more details in it.

Some details were added. See L. 246

L235. It would be worth showing the TKN data on Fig. 2 also to illustrate this point.

The figure 5 of the revised manuscript illustrate the TKN predominant pattern (clockwise).

L240-242. In the methods you introduced the +/-10% cut off for neutral loops, so you just use that here from the outset. So just note that 13% have DR values within the 10% limit and are classified as neutral. As written, it is not very clear how many of the loops you consider to be what type and whether you are being consistent in definitions.

This part of the method and results were modified. See L. 214-218, 262-263, 271-272

L242-244. Not clear, needs rewording.

The sentence was modified

L244-250. Similar comments about the +/-10% apply here and again the text is not very clear.

This was revised. See L. 271-272.

L257. Not sure what "response controls" means.

L258-261. You should discuss that you are using a Pearson correlation matrix and any relevant derails in the methods.

It was indicated in methods.

Discussion

This is a long section and a brief introduction at the start guiding the reader through what you intend to discuss would be good. There is also a tendency to interpret your data by reference to other studies (so on L304-305 you explain the results of others and then discuss your interpretations in the next few sentences). It would be preferable to discuss your interpretations and justify them using your data and then note whether they are similar or different to those elsewhere. Otherwise it looks like you are trying to fit your data into an existing framework, which I don't think is the case.

Following your indications, some parts of the discussion section were written. See L. 330-345, 372-380, 389-405.

L293. Section heading is confusing

The section was modified. See L. 388.

L294-303. This seems to be off topic with the title of this section. Does it belong here? Perhaps it go be fore the sectiion heading and act as a general lead-in?

Your indications were considered. See L. 316-327

L304. What is an "accretion pattern"?

It´s mean increasing, but it was deleted.

L310-321. There is a lot of speculation here (deep drainage, high NO3 in soils, rapid transport). Can you provide justification – there seems to be a number of prior studies on this area that may help.

Some justifications were provided. See L. 330-348.

L324-L330. In section 2, you say that overland flow is unusual, so how can that be?

Surface runoff represents a small portion of streamflow during runoff, but in some cases could be important delivering particulate matter.

L334-350. Similar to the comments above, there is a lot of speculation here. While studies elsewhere may help with that, is there evidence from this catchment that you could use to firm these ideas up.

Some references were added. See L. 70.

L348-351. Not very convincing, what might those sources be?

It was rewritten. See L. 377-379.

L367-370. Very awkward sentence.

The sentence was rewritten. See L. 398-404.

---

## Author Response (AR2)

**Response to reviewers**

In this document, we address the comments and points raised by the reviewers, and we indicate where the changes included in the manuscript to fulfil the comments.

This is a much clearer paper than the original and the authors have answered most of my main questions (Reviewer #2 of the original paper). I have a few minor suggestions on this version.

Thank you for the time you devoted to reading this manuscript. We appreciate the kind comments of Reviewer #2 regarding the revised manuscript (1).

1) Figure 1 is a good addition. However, the top loop in 1A and the bottom loop in 1B both look flat. For illustration purposes it would look clearer if the long axes of those loops were more angled.

The Figure 1 was improved following the reviewer comments. See Figure 1 in the revised manuscript.

2) Figure 2. Would it be beneficial to plot the N concentrations on this plot as well (you could plot then on another panel above the main panel)?

We have followed the reviewer suggesting, including the N concentration in Figure 3. Please see Figure 3.

3) Figure 4. These visualisations are useful but may be better to add those schematic loops to Fig. 6 as that is where the data are plotted and there is ample space in one or both of the parts of Fig. 6 to add those schematics.

Following the reviewer suggestions, the Figures 4 and 6 were merged,

Finally, I suggest moving the final paragraph of the Discussion (L398-404) into the Conclusion and adding it to / merging it with the last paragraph (L413-416) as both deal with overlapping broader implications.
The paragraph was moved to the conclusion section in the revised manuscript. L. 447-450.

Reviewer 2

Dear Reviewer,
Thank you for the time you devoted to reading this manuscript and for your helpful comments. We took your remarks and suggestions into account to improve our manuscript. In addition, the manuscript has been again checked by a native English speaker.

The reviewer points out the lack of event definition and identification and the differences in the statistics used as the weakest aspects of the work. These aspects have been addressed in the corrected version of the work. L.

Specific comments:

Line 17: Not clear what is meant here by the overall dynamics of hysteresis. Please clarify.

Sorry. It was changed in the revised manuscript. We want to say hysteresis direction.

Line 17, 198: The concept of dynamics of hysteresis (dR) is different from the hysteresis direction stated in the Method section. Please use only one term throughout the manuscript.

It was corrected in the revised manuscript. We used hysteresis direction throughout the manuscript.

Line 18: Element is ambiguous, compound is a better term.

Thanks, the term has been changed in the revised manuscript. L. 18.

Lines 20-23: The meaning behind each of the mentioned characteristics (e.g., difference between peak discharge and event magnitude) and the direction of change produced by different event characteristics is not clear at this point. Please use more comprehensive terms.

Thanks. It was clarified in the revised manuscript. L. 20-24.

Line 24: It is not clear what is meant by the magnitude of the event here. Is it the volume or the peak discharge. Please either explain the meaning of the characteristics used in the abstract directly in the abstract or use only unambiguous terms.

The magnitude is (Qmax-Qb)/Qb. It was clarified in the manuscript. L. 22.

Figure 1: Please also label dilation and enrichment pattern. Consider highlighting the line showing the width by a different color. Please clarify in the caption if there is a difference between "rising slope" and "rising limb". If not, use the same term.

The Figure 1 was improved following your comments of both reviewers. Pg. 2

Line 75, 204, 383: do not use the term "significant" if it is not associated with the significance testing.

In agreement. The term has been modified in the revised manuscript following the indications of the reviewer.

Line 106: Please indicate how close the station is.

This information was added in the revised manuscript. L.115-116

Line 110: It is not clear what is provided in the brackets? The range of elevation? Usually, the range is provided as min-max and not the other way around.

Effectively, the information contained in parentheses indicates the elevation range. Sorry for the mistake, in the revised manuscript it has been provided as min-max. L. 101.

Figure 2. Please modify the colors of the land uses, it is difficult to distinguish. Consider

simplifying the land use types. Please indicate the location of the three meteorological stations mentioned in the text,

Figure 2 was improved following the reviewer indications. Pg 5

Lines 133-134: How was the start of the rainfall event defined? To compute the increase relative to the start of the event, the start has to be defined first. Please clarify.

The definition of rainfall event was included in the revise manuscript. L. 133-139.

Lines 139-141: I do not think this information is used in the analysis. Please delete all unnecessary information or clarify how it was used.

All right! The information was deleted, and the paragraph reformulated. L. 144-146

Lines 149-150: This definition of rainfall start does not agree with autosampler operation mode described in Line 133-134. Please also clarify what does "usually exceeds 5 mm" mean? Is there no fixed threshold?

This information was clarified in the revised manuscript. L. 153-159.

Line 153: What is a "perceptible increase"? Is there a clear threshold for that? Please clarify.

This information was clarified in the revised manuscript. L. 153-159.

Line 154: It is not clear if events finish when they reach exactly the same initial baseflow conditions or not. Please clarify. Moreover, given an unclear definition for the event start it is also not clear how exactly the termination of the previous event is defined when it is followed immediately by the next event. Please clarify this too.

This information was clarified in the revised manuscript. L. 153-159.
Line 162: term "water yield" is rather ambiguous, consider using "total runoff volume" instead.
Ok. It was modified following the reviewer indications in the revised manuscript. L. 163; table 1, figure 6.

Lines 164-166 Please clarify how rising and falling limbs are defined. Homogenize usage of Rd and RD.
We clarified how rising and falling limb were defined. L. 168-170. We also included in the Figure 3
Sorry for the mistake. It was corrected in the text of the manuscript and in the table.

Figure 3. Please also display how the slope of the falling limb is defined in this Figure. It would be helpful if all characteristics from Table 1 are displayed in this Figure. Please also make sure that the acronyms in the figure and in the table are identical.
The Figure 3 was improved following the suggestions of both reviewers

Lines 174-175: It is not clear why the number of events has reduced from 156 to 102. Please clarify.

The way in which the events were selected is indicated in the revised manuscript. L. 182-185.

Line 178: Please clarify how close it has to be.

The information was added. L. 183

Table 1: All event characteristics have to be explained in the Table or in the text (e.g., how the initial phase of the falling limb identified?). Please use identical terms for event characteristics in the text and in the table (e.g., event magnitude relative to baseflow is listed here as magnitude of event and can be easily confused with peak discharge).

All right! All event characteristics have been explained in the text.

Line 215: previously the term "direction" was used instead of "rotation". Please keep homogenous terminology throughout the manuscript.

Sorry. We used direction in the revised manuscript.

Lines 217-218: What does it mean? Were these events treated differently? Please clarify.
No, they weren't. It was deleted.

Line 223: which examined characteristics can be considered biogeochemical? Please clarify.

We are taking about nitrogen concentrations. The information was added in the new version of the manuscript. L. 225

Lines 225-227: From this description, it is not clear what is the additional value of the redundancy analysis compared to correlation. Please clarify.

This section was modified; we hope it was improved. L. 226-235.

Line 230-235: a boxplot figure displaying described variability among events will be useful here.
Following the reviewer comments the new figure 4 was added.

Figure 5: I think this figure is more comprehensive than figure 4 and they can be merged together. The date on the d panel is not clear. Please revise.
The data of d panel was corrected.
Following the indications of the other reviewer, the figure 4 was merged with Figure 6

Line 246: Based on the units it should be volume instead of magnitude. It is not clear which data is not shown here. In general, all relevant data should be provided. Moreover, it is not clear what is the general length of recorded rainfall events (min and max).
Sorry for the mistake; it was corrected. Magnitude was replaced by volume. L. 255

It was changed and the min and max values were added. L. 256-257
Line 247: volume instead of magnitude?
You are alright; it is volume. It was changed. L. 257.

Line 251-252: I do not really see it in figure 5. Autumn events are only displayed for NO3 and it is not really much different from winter in terms of concentrations. Moreover, be cautious in drawing such general conclusions from singular cherry-picked examples.

Alright! Now it can be seen in the new Figure 4.
.
Section 3.2. Please add percentage instead of absolute number of events.
Ok, it was added in the revised manuscript. L. 271, 272.

Figure 6: Please, add labels to the subplots. I see no horizontal dotted lines in this figure that are mentioned in the caption. Please clarify.t.

The Figure was improved, now it can be possible to see the horizontal dotted lines. The caption was also completed. L. 289-292

Section 3.3. Please use full names when referring to different event characteristics in the text to make it more readily understandable for the readers.

Ok. The full names were used in the new version of the manuscript. L. 301-325.

Line 305: it is not clear which variables exactly are meant here by "rainfall-runoff magnitude". Please clarify

Ok; it was completed. L. 323-324.
.
Line 330-331: This conclusion is not straightforward. Please elaborate on that.

Line 337-338: This is not a part of the manuscript anymore, you cannot not reference the results that are not presented in the manuscript. Please revise.

Ok. The sentence was reformulated in the revised manuscript. L. 350-351

Line 341-344: This reasoning is not convincing. The authors attribute the anticlockwise hysteresis patterns to different pathways (subsurface and groundwater), however low nitrate concentration in subsurface flow at the beginning of the runoff event could also be linked to the runoff generation zones that are closer to the stream, and therefore with the spatial distribution of sources. Please justify your reasoning on why this is not the case for the study area.

Ok. We have checked the statement and modified it in order to clarify our point. L. 364-369.

Line 356: event water contribution was not quantified in this study. Please revise.
Ok. The paragraph was reformulated. L. 355-360.

Line 359: What is meant here by direct rainfall? Please clarify.

It was clarified in the revised manuscript. L. 383-385

Line 376: the meaning of the information provided in the brackets is not clear. Please elaborate. Please also clarify how the distance of sources is related to event magnitude. Please notice that previously the term event magnitude was used in a different meaning (relative increase of streamflow compared to baseflow, while now it refers to precipitation volume). Please homogenize the terms throughout the manuscript.
The paragraph has been reformulated in order to clarify our point. L. 401-402.

Line 388: which rainfall characteristic is meant here? What is meant here by event magnitude? Please clarify.
 It was clarify. L. 404.

Line 395: This statement needs a reference (e.g., Heathwayte and Bieroza 2021 might be suitable here) and a further explanation on how biogeochemical processes are influencing hysteresis patterns of nitrate.
Thanks. The reference and the explanation were added. L. 424-428.

Line 397: This statement also needs a reference. How does the buildup of nitrate affect the biogeochemical processes involved
Ok. The paragraph was modified in order to clarify the statement. L. 431- 434.

Line 402-403: This sentence has to be revised, it does not make sense.

The sentence was checked and added to the conclusion section following the suggestion of the other reviewer. L. 447-450.

Line 403-404: This is rather vague. What additional event-based studies can provide that was not observed here? Please clarify

Your are right. It was deleted.
 .

Line 412: Here I would expect a summary of main mechanisms that were identified by the dual analysis.
It was added. L. 439-440.

Editorial comments:

Line 18: dominated by enrichmentç
Line 33: freshwater bodies
Line 36: commonly at biweekly or monthly resolution
Line 43: nutrient concentration
Line 48: can be classified

Line 51: distant to the stream
Line 53: omit etc
Line 56: nitrogen species
Line 64: rainfall and runoff event characteristics
Line 78: through
Line 80: changes in the quality of freshwater resources
Line 87: C-Q relationships
Line 100: indicating relatively low nitrogen inputs in the Corbeira catchment compared to…
Line 161: determined according to Wischmeier and Smith (1958)
Line 180: evaluated visually
Figure 4: reduce white space
Line 233: selected events
Line 247: several events
Line 321: the Corbeira
Line 325: increased nitrogen concentration during individual events
Line 338: contribution from the subsurface flow
Line 348: that have recently received fertilizers
Line 383: on hysteresis patterns?
Line 393: particulate material

Sorry for the mistakes. All editorial comments were taken into account and corrected.

References:
Heathwaite, A. L., & Bieroza, M. (2021). Fingerprinting hydrological and biogeochemical drivers of freshwater quality. Hydrological Processes, 35(1), e13973. https://doi.org/10.1002/hyp.13973

Thanks. The reference was added in the revised manuscript.

---

## Author Response (AR3)

**Response to reviewer 1**

In this document, we address the reviewer´s remarks and points, and we indicate where the changes included in the manuscript to fulfil the comments.

"The manuscript is very much improved now… However, some of my comments were not addressed. I still do not see convincing evidences that total nitrogen plays important role in the particular study catchment, as its concentration seems to be much lower compared to the nitrate. Moreover, there is still some ambiguity in event definition and a lack of rationale for selecting particular set of event characteristics for the explorative analysis."

Organic nitrogen (represented in this study by the TKN because $NH_4$ was always found below detection limits) accounts for a moderate fraction (> 25%) of the nitrogen exported from the catchment; its contribution to total nitrogen increases significantly during runoff events (Rodríguez-Blanco et al., 2015). In the revised manuscript, the NO3 concentrations have been shown as $NO_3$-N rather than $NO_3^-$ for ease of comparison. As a result, the graphs and tables have been updated. L. 370-376.

We attempted to complete the characterization of the rain and runoff events by describing how we chose the start and finish points of the runoff. Hopefully, it is now more clear.

Figure 1: Please provide in the caption details on how the width is computed (i.e., at which point).

We included information suggested by reviewer. L. 58-60

Line 73-76: What is a meaningful quantity for total nitrogen? Are the quantities measured in the study catchments are actually "meaningful"? Please clarify.

We've added a paragraph to illustrate the importance of organic nitrogen in many catchments with different land uses.79-85.

Figure 3: Indicate a and b panels. In the panel a it seems like there should be labels for different arrows, but labels are missing. Without the labels the meaning of the arrows is not clear. It is not clear how the starting point is identified (neither from the text nor from this figure). On panel b the meaning of green line is not explained and the line reaches beyond the plot. The meaning of the red line below the legend is not clear. It will be helpful to show two different cases described in the text: single event and event that is immediately followed by the next event, to understand how the cutoff was done.

Indeed, if there is a label; we have re-added the graphic. Hopefully, they may now be appreciated properly. We also changed the graph in panel b and included a new graph depicting a sequence of events as well as the progression of nitrate and TKN concentrations. Please see new Figure 3.

Lines 133-135: This is still not clear. Are these the rules that were applied for autosampler or for further assessment of rainfall events? Does this mean that the autosampler starts to sample no earlier than 10 hours after the end of the previous event? (between event, 10 hours) How the end of the event is defined? The last non-zero rainfall pulse? Does 5 mm rainfall threshold apply to rainfall rates or rainfall volume? Please revise and be more rigorous in your descriptions.

We have revised the text and completed the definition of rainfall and runoff events. L. 141-146.

Rainfall threshold: 5 mm rainfall volume. L. 147.

Line 139: Not clear which hydrological responses are meant here. Please clarify.

It is the hydrological response of the catchment to the rainfall events. L. 149-150.

Lines 150: Please clarify how the beginning of the event is defined.

We attempted to clarify this issue in the revised manuscript. L. 160-166.

Line 154-155: What does unit "cms" stands for? Please explain. What are these three consecutive hydrological records? Do you mean three time steps? Please revise.

The paragraph was modified.

You are right, three time steps.

Line 157: What is meant here by "abrupt change"? Please clarify. Please also explain if the abrupt change is only evaluated for a single time step or only abrupt changes of certain duration are considered? Why an event can only have one abrupt change?

This point, we believe, has already been clarified in the revised manuscript. We examine the change over three time steps (30 minutes). Obviously, there can be more than one abrupt changes, but we're referring to the one at the start of the events. L. 166-166.

Line 158-159: This is not clear. Please revise.

It was modified

Lines 174-178: This is very general and generic description that does not provide any justification about the set of the characteristics selected for the analysis, nor does it clarify which hypotheses the authors were testing in this study.

The paragraph was modified; hopefully it´s clearer now. L. 170-176

Table 1. The values of total runoff look erroneous. Please revise. Sorry for the mistake; it was corrected. The values were in $m^3$. Please see the table 1.

Equation 2: It is not clear why the authors chose to use different denominator here. Butturini et al 2006 only used Cmax for normalization. Please clarify.

Sorry, its Butturini et al. 2008 not 2006. It was corrected.

Figure 4: The definition of the boxplots is unclear. Please revise. Provide the number of events for each season on top of the corresponding box.

Following the reviewer comment, the caption and the figure were completed in the revised manuscript. L. 267-271.

Table 2: The correlation between Cinitial, Cmean and Cmax is quite high with the deltaC, but these concentrations are actually used to compute deltaC (equation 2). Does it make sense to evaluate their correlations?

Certainly, deltaC is calculated using Cmax and Cinitial. However, we believe that the corrections help to demonstrate the differences in the behavior of $NO_3$ and TKN concentration. For this reason, we chose not to remove them from the manuscript.

Figure 7: Please explain the caption the meaning of the axes, and vectors to help the readers to interpret the results presented in this Figure.

The figure caption was completed according to the reviewer's specifications. We also had to change the figure after discovering a mistake in it. L. 343-348.

Editorial comments:

Line 12: not clear what is meant here by "from streams".

It was corrected. L. 12

Line 16: clockwise, anticlockwise and no hysteresis.

It was corrected. L. 16

Line 20-26: acronyms can be avoided in the abstract.

The acronyms were deleted.

Line 35: these countries

It was added. L. 35

Line 44-46: Please revise this sentence, it is not clear.

It was revised. L. 44-46

Line 53: spatio-temporal

It was corrected. L. 53

Line 109-110: keep units on the same line

It was corrected.

Figure 2: catchment boundary and railroad have identical symbols.

The catchment boundary was modified

Line 181: a runoff event

It was corrected

Line 401: clarify which pattern is meant here

It was clarified. L. 431